# When Stability meets Sufficiency: Informative Explanations that do not Overwhelm

**Ronny Luss**      *rluss@us.ibm.com*
*IBM Research, Yorktown Heights*

**Amit Dhurandhar**      *adhuran@us.ibm.com*
*IBM Research, Yorktown Heights*

**Reviewed on OpenReview:** *https://openreview.net/forum?id=8JNXOB6FtW*

## Abstract

Recent studies evaluating various criteria for explainable artificial intelligence (XAI) suggest that fidelity, stability, and comprehensibility are among the most important metrics considered by users of AI across a diverse collection of usage contexts. We consider these criteria as applied to feature-based attribution methods, which are amongst the most prevalent in XAI literature. Going beyond standard correlation, methods have been proposed that highlight what should be minimally sufficient to justify the classification of an input (viz. pertinent positives). While minimal sufficiency is an attractive property akin to comprehensibility, the resulting explanations are often too sparse for a human to understand and evaluate the local behavior of the model. To overcome these limitations, we incorporate the criteria of stability and fidelity and propose a novel method called *Path-Sufficient Explanations Method (PSEM)* that outputs a sequence of stable and sufficient explanations for a given input of strictly decreasing size (or value) – from original input to a minimally sufficient explanation – which can be thought to trace the local boundary of the model in a stable manner, thus providing better intuition about the local model behavior for the specific input. We validate these claims, both qualitatively and quantitatively, with experiments that show the benefit of PSEM across three modalities (image, tabular and text) as well as versus other path explanations. A user study depicts the strength of the method in communicating the local behavior, where (many) users are able to correctly determine the prediction made by a model.

## 1 Introduction

Consider an approved loan applicant that asks their bank the following question: "How can I change my profile and still remain in good standing?" This is a common occurrence, for example, when someone wants to make a major purchase, e.g. buy a car, after their mortgage is approved but before the home purchase is officially completed (see the article "What is Considered a Big Purchase During Mortgage Underwriting" (Sharkey, 2022)), when a married couple divorces and each separate party needs to obtain a new mortgage while negotiating a settlement on their joint finances, or when parents want to relocate after a child graduates high school while simultaneously paying for college; they all need to understand how the change to their finances impacts their credit. The applicant is essentially asking how much, if any, additional debt/risk can be taken without affecting the desired mortgage. One potential answer, called the minimally sufficient explanation or *pertinent positive* (PP) (Dhurandhar et al., 2018; Feng et al., 2018), highlights the minimal set of features (and values) that are sufficient to replicate a model's decision, which is the minimal applicant profile that will cause the model to approve the loan[1]. In the college scenario, the parents want to compute how much

---

[1]This is complimentary to counterfactual explanations which answer the opposite question "Which features do I need to change in order to become a good applicant?" asked by a rejected applicant.

**Table 1:** Stable path of scenarios where applicants remain approved for loans in the HELOC FICO (2018) dataset. The model being explained is a one layer neural network trained to 72% accuracy on classifying good vs bad credit standing. External Safety Estimate is a metric where higher safety is better. Each column is a reduction in the Original values that maintain the model's prediction of loan approval, with PSEM 1-4 being our algorithm's monotonically decreasing path of applicant profiles versus CEM-PP, which is a much less realistic profile (e.g. opening many new accounts at the same time, observed because Age of Newest Account is 0, in order to maintain the same total number of accounts is not practical).

**Applicant 1**

| Significant Variables | Original | PSEM-1 | PSEM-2 | PSEM-3 | PSEM-4 | CEM-PP |
|---|---|---|---|---|---|---|
| External Safety Estimate | 84.0 | 81.9 | 81.9 | 81.9 | 81.9 | 84.0 |
| Age of Oldest Account | 341.0 | 279.2 | 256.9 | 250.5 | 250.5 | 42.5 |
| Age of Newest Account | 11.0 | 11.0 | 11.0 | 10.9 | 8.6 | 0.0 |
| % Accounts. Never Delinquent | 100.0 | 89.5 | 87.1 | 86.8 | 86.8 | 100.0 |
| Total # of Accounts | 11.0 | 7.5 | 6.0 | 5.5 | 4.3 | 11.0 |

**Applicant 2**

| Significant Variables | Original | PSEM-1 | PSEM-2 | PSEM-3 | PSEM-4 | CEM-PP |
|---|---|---|---|---|---|---|
| External Safety Estimate | 88.0 | 79.6 | 79.6 | 79.6 | 79.6 | 88.0 |
| Age of Oldest Account | 170.0 | 139.2 | 135.5 | 127.2 | 127.2 | 13.6 |
| Age of Newest Account | 3.0 | 3.0 | 3.0 | 0.6 | 0.4 | 0.0 |
| % Accounts Never Delinquent | 100.0 | 86.8 | 86.8 | 86.8 | 86.8 | 92.7 |
| Total # of Accounts | 15.0 | 11.7 | 11.0 | 9.7 | 7.0 | 15.0 |

cash they can contribute towards tuition (which reduces any required college loans) while maintaining good credit standing for a new mortgage.

While such an explanation might answer the applicant's question, the highlighted features are often too few to act on in a practical manner (e.g., a true minimal solution does not contain enough explanatory/actionable information). Rather, a stable path of solutions will offer the applicant a better perspective as seen in Table 1 which answers the question about remaining in good standing to two loan applicants from the Home Equity Line of Credit (HELOC) dataset. Our measure of stability, illustrated below in Figure 1 and formally defined in Section 3, connects each solution along the path, in effect forcing the applicant profiles to remain realistic. The HELOC dataset FICO (2018) contains credit applicant data, describing various aspects of an applicant's credit history such as how many credit lines s/he has open and various features about delinquent payments, where the goal is to predict whether or not someone is a good applicant.

The Original column is the applicant's current profile, and the columns labeled PSEM 1-4 offer a stable path of reducing profile feature values while maintaining an approval rating by the bank's model. For example, it shows at what rate the applicant can close accounts while also reducing their estimate of being a safe applicant (External Safety Estimate) or reducing the percentage of accounts that have never been delinquent. If the applicant was only given the perspective of the PP (termed CEM-PP in the last column), it would seem the applicants must close many old accounts (reduce Age of Oldest Account), and open several new accounts (reduce Age of Newest Account to 0) in order to maintain the same number of original accounts, which is impractical and against intuition. This example motivates the need for stable paths of sufficient explanations that lead to better insight, which is the main contribution of this work.

Utilization and impact of black-box models (e.g., neural networks) has significantly grown creating the need for new tools to help users understand and trust models (Gunning, 2017). Given this acute need, various methods have been proposed to explain local decisions of classifiers. We seek to consider local explanations from the perspectives of stability, fidelity, and comprehensibility - these are among the most important criteria considered by users of AI across a diverse collection of usage contexts (Liao et al., 2022). Most such local methods highlight relevant features that determine the model's decision, but we focus on methods that highlight the PP as described above that typically involve some type of control over the sparsity of the explanation. One could develop a path of explanations by varying such regularization, but we find such paths often lack stability which is crucial for building trust in the explanation.

**Figure 1:** First column is the original image. Masks are used to show visible facial features as explanations. Sparsity increases left to right. PSEM shows stable paths of sufficient explanations, whereas CEM-PP exhibits instability, as higher sparsity settings sometimes result in denser explanations.

This instability is illustrated in Figure 1 which compares a path of PPs (termed CEM-PP) with our method termed PSEM. Original images are in the first column, each of which is followed by a path of explanations. Notice that, while the PSEM path offers sequential explanations each of which is a subset of the previous explanation, a property which we call stable, column 4 of CEM-PP is denser than column 3 meaning that a higher sparsity level led to a less insightful explanation, which is counterintuitive. PSEM is designed to offer path explanations that lack such instabilities.

Moreover, for a particular PP, one may not even be able to find a "close enough" sufficient example that is not the entire input. This means that while the PP offers the smallest sufficient set of features to maintain the classification, there may not exist a "second smallest" sufficient set of features (other than the entire input) to maintain the classification. Since the PP might be too sparse for a human to interpret, perhaps a slightly larger sufficient example would be preferable; however, this slightly larger sufficient example may not exist deeming the explanation not comprehensible since it cannot be related to slightly more complex explanations. Hence, a stable sufficient path is important to give comprehensibility, since one can see which features maintain sufficiency while not letting the corresponding explanations cross over to classes different from the original input (e.g., PSEM solutions also maintain fidelity). We hypothesize that the stability and fidelity of our path explanations leads to better assessment of model predictions, i.e., are more comprehensible. This hypothesis is validated in experiments, both quantitatively and qualitatively, and through a user study. From the quantitative perspective, our explanations outperform competing path methods in terms of fidelity and stability, two metrics that have been shown to be most important to users of explainability across multiple contexts (Liao et al., 2022), implying that users would have greater trust in our explanations.

The user study serves a second key intention as it is a priori not obvious if extra information helps or overwhelms users. In fact, Hancox-Li (2020) argue that explanations should be smooth and robust as users are often more interested in *identifying persistent patterns* in the data rather than focusing on quirks of the model. Our results validate this need to offer users more information in explanations, both to build user trust (as measured in the user study) and to give consistent insights (as measured quantitatively). Interesting observations are made; for example, users with a preference for less informative PP explanations unknowingly performed better on the user study task given more informative PSEM explanations.

Our formulation in Section 3 instills stability and fidelity along the PSEM path and is the main novelty; to the authors' knowledge, no other explainability work consider paths for attribution methods. Typically, attribution methods are tuned to adhere to some principle, be it sufficiency as in PPs or relevancy as in LIME (Ribeiro et al., 2016), and in the case of PPs, we demonstrate that a path of PPs can offer greater insight, and open the door to other potential path explanations. Our contributions are summarized as:

1. We propose a novel (constrained) formulation to learn a stable sequence of sufficient explanations.

2. We propose methods to efficiently solve the optimization problems using custom alternating minimization.

3. We quantitatively demonstrate the benefits of our path explanations by adapting known posthoc explanation methods.

4. While previous sufficiency-based methods were applied to particular modalities, PSEM is applied to three.

5. We show the value of our method through a standard task-based user study that tests explanation understanding.

## 2    Related Work

Identifying important features of an input that are relevant to its classification has been an active research area (Zhou et al., 2021; Ribeiro et al., 2016; Lundberg & Lee, 2017; Simonyan et al., 2013; Lapuschkin et al., 2016). An attribution method (Schulz et al., 2020) inspired by the information bottleneck (N. Tishby & Bialek, 1999) quantifies the amount of information each image attribute contributes to a prediction. Several works pursue explanations that require data to train neural networks for providing explanations. Situ et al. (2021) pursues stable explanations by learning a model that predicts the explanations given by an input explanation model such as (Ribeiro et al., 2016; Zeiler & Fergus, 2014), Yoon et al. (2019) trains predictor, baseline and selector neural networks to offer explanations from the selector network, Jethani et al. (2019) trains a global selector network to identify the important features, and Dabkowski & Gal (2017) focuses on image classifiers and trains a large neural network for inferring image masks. These explanation methods all differ from the scenario considered in this paper where we do not assume sufficient data to train additional neural networks. Another attribution method by Zhou et al. (2015) is tailored to explain convolutional neural networks, particularly focuses on pooling operations, and is not a general explanation method. Lastly, Zhou et al. (2021) uses hypothesis testing to decide when more samples are necessary to induce stability.

Hierarchical explanations (Singh et al., 2019; Tsang et al., 2018) offer a path of explanations that highlight which feature interactions are most meaningful to the prediction, i.e., which feature interactions flip the classification to the predicted class from some other class. While informative, such explanations offer a different type of information than considered here, more in line with counterfactuals (Guidotti et al., 2018; Le et al., 2020) or the pertinent negatives defined in Dhurandhar et al. (2018). Pertinent positives rather offer a complimentary explanation to such methods.

Most relevant to our paper are local sufficiency based methods (Dhurandhar et al., 2018; Feng et al., 2018; Ribeiro et al., 2018; Lei et al., 2016; Luss et al., 2021; Fong et al., 2019; Carter et al., 2019; 2021; Watson et al., 2021) which highlight features that are sufficient to obtain the same classification as the original input. The explanations provided by these methods may be too sparse to judge the quality of the model. Moreover, a higher sparsity penalty may not always lead to a sparser explanation. Masking and infilling has been used in previous sufficiency-based explanations for colored images such as Fong & Vedaldi (2017) and Chang et al. (2019), where the main difference is on how infilling is done. Luss et al. (2021) and Fong et al. (2019), which builds on Fong & Vedaldi (2017), consider smoother masks than these prior works; Fong et al. (2019) considers smooth masks over individual pixels while Luss et al. (2021) consider smooth masks according to a prior image segmentation. These different perturbation-based methods are extendable to our framework, as will be demonstrated on a colored image dataset specifically for the method of Luss et al. (2021). Lei et al. (2016) jointly trains a classifier with an explanation generator for text classification; we explain fixed models. Carter et al. (2019) is a form of input reduction (Feng et al., 2018), and the very recent extension to Carter et al. (2021) uses gradients to select features rather than expensive greedy search. Watson et al. (2021) offers a probabilistic measure of sufficiency by sampling and enumerates over all possible sufficient explanations and is impractical in high dimensions. Lastly are local sufficient explanation methods (also called abductive explanations or prime implicants) for logic-based models such as decision graphs (Ignatiev et al., 2021; Darwiche & Ji, 2022). While our focus is on complex differentiable models (viz. deep neural networks), there is a large body of literature focused on these logic-based models, where a sufficient explanation seeks a minimal set of features that matter for a prediction, whereas our framework allows features to simply be reduced, such as in the loan approval example in the Introduction.

# 3  Problem Statement and Method

We start by introducing a formulation for learning a path of sufficient explanations and relate it to finding PPs in Dhurandhar et al. (2018). Let $x_0$ denote an input with output label $t_0$ and $\mathcal{X}$ the entire input space. For the different modalities considered, we assume $\mathcal{X} = [0,1]^n$ for text documents with $n$-word normalized dictionaries or grayscale images with $n$ pixels, and $\mathcal{X} \subseteq \mathcal{R}_+^n$ for tabular data with $n$ features, i.e., tabular data that has only non-negative features. Let $\text{Pred}(\cdot)$ denote a function that takes an input and outputs the predicted probability vector for a classifier. $[\text{Pred}(\cdot)]_i$ denotes the $i^{th}$ component of this vector, i.e., the probability of being classified in the $i^{th}$ class. A path of $N$ explanations is denoted by $\boldsymbol{\delta}_0, \ldots, \boldsymbol{\delta}_N \in \mathcal{X}$ and can be viewed as samples that lie in the same space as input sample $x_0$. Define the loss function $f_\kappa(\mathbf{x}_0, \boldsymbol{\delta}) = \max\{\max_{i \neq t_0}[\text{Pred}(\boldsymbol{\delta})]_i - [\text{Pred}(\boldsymbol{\delta})]_{t_0}, -\kappa\}$ with margin parameter $\kappa \geq 0$ which applies a penalty if $\delta$ is classified different from $x_0$ (or has a similar score within the margin $\kappa$). Our new formulation, which we call the Path-Sufficient Explanation Problem, is as follows:

$$\min_{\boldsymbol{\delta}_0, \ldots, \boldsymbol{\delta}_N} \quad \sum_{i=1}^{N} c \cdot f_\kappa(\mathbf{x}_0, \boldsymbol{\delta}_i) + \beta \sum_{i=1}^{N} \|\boldsymbol{\delta}_i\|_1 \qquad \text{subject to} \tag{1}$$

$$\boldsymbol{\delta}_0 = x_0, \quad \boldsymbol{\delta}_i \in \mathcal{X} \tag{a}$$

$$\|\boldsymbol{\delta}_i - \boldsymbol{\delta}_{i-1}\|_2^2 \leq \epsilon, \quad \boldsymbol{\delta}_i \leq \boldsymbol{\delta}_{i-1} \quad \forall i \in [N] \tag{b}$$

where $N \in \mathcal{N}$ is a hyperparameter defining the length of a path that must be tuned for a particular dataset. We use notation $[N] = \{1, \ldots, N\}$. The first term in the loss ensures that each explanation along the path maintains the same class as the input, i.e., fidelity. The second term in the loss penalizes the number of features in successive explanations. The constraint in (a) defines $N$ variables $\boldsymbol{\delta}_i$ that represent the sufficient explanations we learn along the path, where the first explanation is the input sample $x_0$ and the last explanation is the remaining features that appear in the objective's loss function. The second constraint in (a) represents feasibility for the domain. The first constraints in (b) enforce stability (each successive explanation in the path is close to the previous one); El Zini & Awad (2022) call such constraints the $\epsilon$-connectedness of $\delta_0$ to $\delta_N$ which they used as a metric for faithfulness of counterfactuals. The second constraints in (b) enforce monotonicity of the explanations along the path; successive explanations highlight only a subset of the features highlighted in prior explanations, where $\boldsymbol{\delta}_i \leq \boldsymbol{\delta}_{i-1}$ is defined component-wise. This gives comprehensibility to the path as features not highlighted before do not suddenly start showing up as important, which can be intuitively hard to understand. Taken together these constraints inform as to what it might take to remain in the particular class as we drop or reduce importance of features.

The constraints in (b) are the main innovation over Dhurandhar et al. (2018), as given a PP solution $\boldsymbol{\delta}^*$ from Dhurandhar et al. (2018), there is no guarantee that a $\hat{\boldsymbol{\delta}}$ exists with $\arg\max_j[\text{Pred}(\boldsymbol{\delta}^*)]_j = \arg\max_j[\text{Pred}(\hat{\boldsymbol{\delta}})]_j$ such that $\|\boldsymbol{\delta}^* - \hat{\boldsymbol{\delta}}\|_2^2 \leq \epsilon$, $\hat{\boldsymbol{\delta}} \in \mathcal{X}$, and $\boldsymbol{\delta}^* \leq \hat{\boldsymbol{\delta}}$ for some small $\epsilon > 0$. Essentially, this means that there are no small additions within the boundaries of the original sample that could be made to the PP that would maintain the same classification. In other words, it implies a lack of stability where judging the quality of the model just based on $\boldsymbol{\delta}^*$ might be challenging, i.e., not comprehensible. This however does not imply that a stable path cannot exist, which motivates the Path-Sufficient Explanation Problem (1). Hyperparameters $c$, $\beta$, $\epsilon$ are to be tuned (see Appendix).

Regarding feasibility, taking $\boldsymbol{\delta}_i = x_0$ for all $i$ is a feasible, but clearly suboptimal, solution since the sparsity regularization in the objective could be reduced. There may be paths where the solution requires $\boldsymbol{\delta}_j = \boldsymbol{\delta}_i$ for all $j \geq i$ for some $i$ along the path, meaning a sparser solution along the path at some point cannot be found, but such a solution resolves the stability issue lacking in Dhurandhar et al. (2018).

One way to solve problem (1) is alternating minimization where each iteration executes the following steps: for $i = 1 \ldots, N$, optimize problem (1) over $\delta_i$ while $\delta_j$ for $j \neq i$ are fixed. In this algorithm, $\delta_i$ is constrained to be within $\epsilon$ of both $\delta_{i-1}$ and $\delta_{i+1}$ for $1 < i < N$, but the constraint $\|\delta_i - \delta_{i+1}\|_2^2 \leq \epsilon$ can be removed because it will be enforced at iteration $i + 1$. Regularization is used to approximate the solution to the resulting constrained optimization.

---

**Algorithm 1** Path-Sufficient Explanations Method (PSEM)

---

1: **Input:** example $(x_0, t_0)$, neural network model Pred$(\cdot)$, $c$, $\beta_i$, $\eta$, $N$, and if applied to color images, a set of binary masks created from a segmentation
2: **for** $i = 1$ to $N$ **do**
3:    If data is tabular, grayscale image, or text, solve:

$$\min_{\{\boldsymbol{\delta} \in \mathcal{X}, \boldsymbol{\delta} \leq \boldsymbol{\delta}^*_{i-1}\}} c \cdot f_\kappa(\mathbf{x}_0, \boldsymbol{\delta}) + \beta_i \|\boldsymbol{\delta}\|_1 + \eta \|\boldsymbol{\delta} - \boldsymbol{\delta}^*_{i-1}\|_2^2. \tag{2}$$

4:    Else if data is a color image:

$$\min_{\boldsymbol{\delta} \in \mathcal{M}(\boldsymbol{\delta}^*_{i-1})} c \cdot f_\kappa(\mathbf{x}_0, \boldsymbol{\delta}) + \beta \|M_{\boldsymbol{\delta}}\|_1 + \eta \|M_{\boldsymbol{\delta}} - M_{\boldsymbol{\delta}^*_{i-1}}\|_2^2 \tag{3}$$

   by prox minimization to obtain $\delta^*_i$
5: **end for**
6: Return $\delta^*_1, \ldots, \delta^*_N$

---

This leads to an algorithm for learning a sequence of explanations, formalized in Algorithm 1, which we call the Path-Sufficient Explanations Method (PSEM). At each iteration $i$, a successive explanation is learned by solving problem (2) in Algorithm 1. $N$ iterations corresponds to executing one iteration of alternating minimization to solve a regularized approximation of the Path-Sufficient Explanation Problem (1). Problem (2) is solved via a prox-algorithm (Beck & Teboulle, 2009) that iteratively minimizes $g(\boldsymbol{\delta}) + \beta_i \|\boldsymbol{\delta}\|_1$ where $g(\boldsymbol{\delta})$ is a first-order approximation to the remaining terms of the objective. Computationally, PSEM requires exactly $N$ times the cost of producing a single PP explanation (Dhurandhar et al., 2018; Luss et al., 2021), where $N$ is the PSEM path length (typically $< 10$). In our setting, this means we are applying FISTA (Beck & Teboulle, 2009) to a nonconvex problem which is analyzed in Li et al. (2017). Note that parameter $\beta_i$ is indexed as we want sparsity to increase through the sequence.

In order to apply PSEM to color images, we adapt an extension of CEM-PP to color images (Luss et al., 2021) in a similar manner as described above. This extension of CEM-PP to color images notes that explanations for color images require a new perspective because, while adding or removing features from grayscale images simply reduces to increasing and decreasing pixel intensities, we cannot similarly optimize over pixel intensity in RGB space as there is no concept of background and the results would be unrealistic. Rather, the concept of feature removal is done by masking superpixels (segments composed of multiple pixels) that result from an image segmentation. Hence, rather than optimize over an input space $\mathcal{X}$, we obtain PPs for color images by optimizing over the space of images resulting from any possible binary mask applied to the input image. The binary masks are predetermined by a fixed segmentation of image $\boldsymbol{\delta}$, and this space of masked images is denoted by $\mathcal{M}(\boldsymbol{\delta})$. Let $M_\delta$ denote the binary mask that when applied to input image $x_0$ produces image $\delta$. Then the PSEM algorithm for color images solves, at each iteration, problem (3) in Algorithm 1.

In practice, the following steps are taken: 1) $\boldsymbol{\delta}^*_{i-1}$ is segmented, 2) an individual mask is created per segment, and 3) we optimize over binary variables (relaxed to lie in $[0, 1]$, constrained to be less than or equal to those that define $M_{\boldsymbol{\delta}^*_{i-1}}$, and then thresholded) that turn individual masks on/off creating a mask in $\mathcal{M}(\boldsymbol{\delta}^*_{i-1})$.

It is also important to consider that the solution to problem (1) is not unique, as the solutions to the subproblems (2) and (3) are not unique. In the event that multiple equivalent global minima exist, the initial point will determine which minima is learned. With that in mind, it is also important to recall that the objective is non-convex with multiple local minima, and so even if there exist multiple global minima, a local minima may be learned instead due to the initial point.

## 4   Quantitative Evaluations

In this section, we provide further motivation for PSEM in terms of the stability and fidelity that are maintained as compared with other path explanations. PSEM is compared with three state-of-the-art methods,

**Table 2:** Results comparing stability and fidelity of path methods on CelebA. Stability measures the % of features at each index ($> 1$) along a path that also appeared at the previous index. Fidelity measures the % of predictions at each index that are equal to the original prediction. Metrics are averaged over 95 samples with one standard error shown in parentheses.

| | Explanation Method | Index Along the Path | | | |
| | | 1 | 2 | 3 | 4 |
|---|---|---|---|---|---|
| # Features | ALIME | 157 (1.1) | 117 (3.3) | 40.8 (2.9) | 8.8 (0.9) |
| # Features | LIME | 162 (1.0) | 68.4 (2.1) | 21.1 (1.3) | 5.6 (0.6) |
| # Features | CEM-PP | 44.9 (3.0) | 24.0 (2.8) | 22.7 (2.8) | 21.5 (3.7) |
| # Features | PSEM | 149 (1.5) | 68.8 (1.4) | 27.8 (2.1) | 23.5 (4.0) |
| Stability | ALIME | – | 95.9 (0.2) | 96.8 (0.3) | 99.3 (0.3) |
| Stability | LIME | – | 97.8 (0.2) | 85.4 (1.0) | 95.0 (1.0) |
| Stability | CEM-PP | – | 62.2 (1.6) | 57.1 (2.3) | 47.9 (3.0) |
| Stability | PSEM | – | 100 (0.0) | 100 (0.0) | 100 (0.0) |
| Fidelity | ALIME | 51.1 (5.2) | 48.9 (5.2) | 42.4 (5.2) | 33.7 (4.9) |
| Fidelity | LIME | 52.1 (5.1) | 47.9 (5.1) | 38.3 (5.0) | 26.6 (4.6) |
| Fidelity | CEM-PP | 100 (0.0) | 100 (0.0) | 100 (0.0) | 100 (0.0) |
| Fidelity | PSEM | 100 (0.0) | 100 (0.0) | 100 (0.0) | 100 (0.0) |

CEM-PP (Dhurandhar et al., 2018), LIME (Ribeiro et al., 2016), and our more stable version of LIME that we term ALIME. ALIME is based on averaging LIME explanations and increases stability along the path as seen in Table 2. While CEM-PP, LIME, and ALIME are meant to be used as individual explanations in practice, one could compute paths of CEM-PP, LIME, and ALIME explanations by varying regularization parameters. Regarding the choice of comparisons, note that CEM-PP is the most relevant and LIME (and its variants) is the most common explanation tool used in XAI literature. The other most popular attribution method today is SHAP (Lundberg & Lee, 2017), but there is no natural way to control sparsity with SHAP as there is with LIME. As discussed earlier, path explanations that vary regularization parameters may find irrelevant yet sufficient artifacts of the classifer (CEM-PP) or insufficient explanations that yield different predictions than the original prediction (LIME). PSEM rather finds less artifacts and always finds sufficient explanations (per the definition of PSEM).

These findings are illustrated in Table 2, which can also be considered an ablation over sparsity parameters for learning the path. Paths of LIME and ALIME explanations were generated using a linear regression with ElasticNet (Zou & Hastie, 2005) regularization for local approximations and varying the sparsity parameter along the path. CEM-PP path explanations were generated by varying $\beta$ in problem (2). Three metrics are considered: 1) the number of features used in each explanation along the path, 2) stability which is the percentage of features used in an explanation that were also used in the previous explanation along the path (hence is undefined at index 1), and 3) fidelity which is the percentage of explanations that maintain the original prediction. Stability less than 100%, referred to as instability, means that features not previously used in an explanation are used in a likely sparser explanation (which is the quality that leads to finding irrelevant artifacts of a classifier). This definition of stability pertains specifically to path explanations such as PSEM or those described above that vary regularization parameters; another definition of stability that typically applies to individual explanations measures whether similar samples have similar explanations. Fidelity less than 100%, referred to as infidelity, means that features used for explanations do not always yield the expected prediction implying that one may question whether an explanation is to be trusted.

LIME, ALIME, and PSEM provide more stable paths than CEM-PP in terms of the number of features; indeed, CEM-PP provides more erratic solutions as the regularization parameter is decreased, while PSEM offers better control along the path due to stability parameter $\eta$. Stability and fidelity for PSEM and fidelity for CEM-PP are 100% by definition. Instability of the CEM-PP path observed in Figure 1 in the Introduction is shown here quantitatively, which is worse than the instability of LIME due to erratic solutions found along the path. Infidelity of LIME is more significant than instability and implies that validity of the path may be questioned. In particular, observe that LIME achieves a lower number of features than CEM and PSEM at the cost of very low fidelity which is consistently among the metrics of greatest importance to users of explainability across various contexts (Liao et al., 2022). ALIME offers improved stability over

LIME along with slightly better, but still poor, fidelity, at the cost of interpretability (i.e., more selected features). Overall, PSEM features patterns of stability that are not always achieved by LIME and even less so by CEM-PP. For these reasons, remaining experiments will be restricted to LIME rather than its variants. See the Appendix for metrics on another dataset, where improved stability of ALIME comes at the cost of fidelity rather than interpretability, i.e. improved stability with fewer features but reduced fidelity for (A)LIME.

**Document 1** (Correct: sci.electronics, Model: sci.electronics)

| | |
|---|---|
| Much deleted about assembly in USA vs. other, I wish to focus on the subject of warm-running amplifiers: . . . a positive correlation with low distortion and "good" sound quality, and high bias results in warmer operation, . . . | |
| **Explanation** | **Selected Words** |
| PSEM-1 | 'amplifi', 'assert', 'audibl', 'bear', 'beleiv', 'bia', 'combin', 'correl', 'current', 'degrad', 'detriment', 'draw', 'driver', 'effect', 'equal', 'equip', 'factor', 'focu', 'good', 'hour', 'idl', 'junction', 'lifespan', 'linear', 'listen', 'mani', 'minut', 'much', 'qualiti', 'resist', 'run', 'stage', 'sucept', 'temp', 'temperatur', 'usa', 'variat', 'warmer' |
| PSEM-2 | 'amplifi', 'audibl', 'bia', 'current', 'detriment', 'good', 'lifespan', 'listen', 'temperatur', 'warmer' |
| PSEM-(3,4) | 'amplifi', 'bia', 'temperatur' |
| CEM-PP | 'amp', 'assembl', 'bear', 'beleiv', 'circuit', 'detriment', 'devic', 'equal', 'idl', 'junction', 'lifespan', 'output','qualiti', 'resist', 'run', 'stage', 'switch', 'vari', 'variat' |
| Input Red. | 'amplifi', 'circuit', 'resist', 'temperatur' |
| LIME | 'temperatur(+)', 'circuit(+)', 'sucept(-)', 'amp(+)', 'driver(-)','low(+)', 'conclus(-)', 'resist(+)', 'linear(-)', 'resist(+)' |

**Document 2** (Correct; sci.med, Model: rec.autos)

| | |
|---|---|
| I use a ZYGON Mind Machine as bought in the USA last year. Although it's no wonder cure for what . . . suppose you're tired and want to go to bed/sleep. BUT . . . I slip on the Zygon and select a soothing pattern of light & sound, and quickly I just can't concentrate on the previous stuff. Your brain's cache kinda get's flushed, and . . . | |
| **Explanation** | **Selected Words** |
| PSEM-1 | 'addit', 'ail', 'bed', 'bought', 'cach', 'churn', 'enhanc', 'kinda', 'light', 'mind', 'new', 'next', 'overal', 'player','quickli', 'select', 'slip', 'sound', 'strang', 'stuff', 'tape', 'tire', 'unresolv', 'use', 'wonder' |
| PSEM-2 | 'ail', 'bed', 'bought', 'churn', 'kinda', 'new', 'quickli', 'slip', 'sound', 'stuff', 'unresolv', 'use' |
| PSEM-3 | 'ail', 'bought', 'churn', 'new', 'quickli', 'slip', 'sound', 'stuff', 'unresolv', 'use' |
| PSEM-4 | 'bought', 'new', 'quickli', 'slip', 'sound', 'stuff', 'use' |
| CEM-PP | 'addit', 'age', 'ail', 'bought', 'cach', 'effect', 'enhanc', 'light', 'machin', 'mind', 'next', 'overal', 'player', 'slip', 'sound', 'tape', 'tire', 'use' |
| Input Red. | **NO WORDS SELECTED** |
| LIME | 'facil(-)', 'elev(-)', 'bought(+)', 'tire(+)', 'stuff(+)', 'slip(+)', 'addit(+)', 'pill(-)', 'surfac(-)', 'quickli(+)' |

**Table 3:** Explanations for one correctly and one incorrectly classified 20 Newsgroups documents.

## 5 Qualitative Evaluations

We next illustrate the usefulness of PSEM on text and image datasets (the introduction already demonstrated PSEM on the tabular HELOC dataset). Each experiment compares PSEM with CEM-PP and LIME (and Input Reduction (IR) (Feng et al., 2018) on text). Among other notable methods, Anchors (Ribeiro et al., 2018) has been applied only to text and the information-theoretic attribution method (Schulz et al., 2020) only to images. Sufficient input subsets (Carter et al., 2019) outputs identical solutions to input reduction. Implementation details, along with a hyperparameter discussion, can be found in the Appendix.

### 5.1 Text data: 20 Newsgroups

The 20 Newsgroups dataset contains text documents labelled as one of twenty categories from different topics among computers (graphics, hardware, etc.), recreation (autos, baseball, etc.), science (electronics, medicine, etc.), politics (guns, mideast, miscellaneous), and religion. Headers, signature blocks, and quotation blocks have been removed from each document, which is processed as a bag of words (all words are stemmed and stopwords removed) with TFIDF normalization. We trained a three layer neural network that achieves 64% test accuracy. Recall that our goal is to explain model predictions, not to build a state-of-the-art model.

We compare PSEM, CEM-PP, LIME, and IR (Feng et al., 2018), a greedy method that removes words, one by one, as long as the classification does not change. For LIME, we set the number of features to 10 for

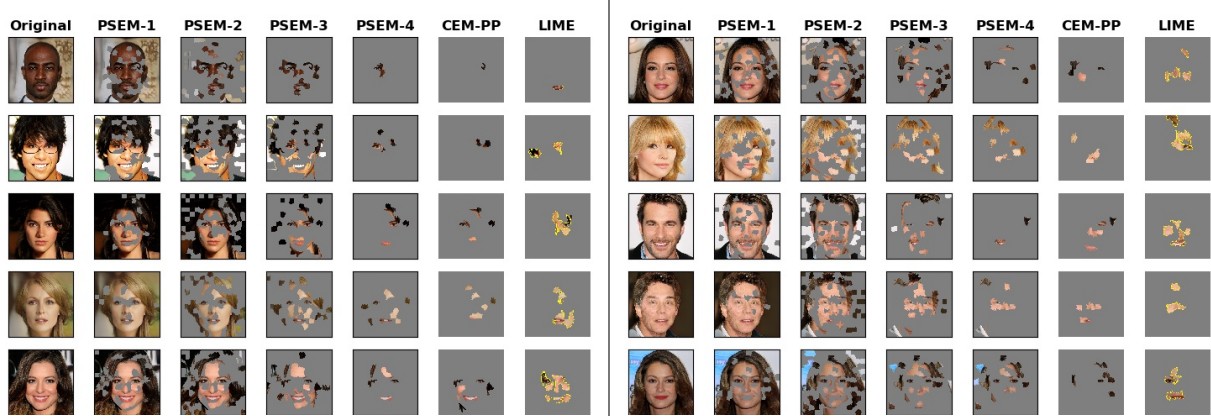

**Figure 2:** PSEM, CEM-PP, and LIME compared on CelebA dataset. Classes of the first five images (left, top to bottom) are (1) Young, Male, No Smile, (2) Young, Male, Smile, (3) Young, Female, No Smile, (4) Young, Female, No Smile, (5) Young, Female, Smile. Classes of the second five images (right, top to bottom) are are (1) Young, Female, Smile, (2) Young, Female, No Smile, (3) Young, Male, Smile, (4) Old, Male, No Smile, (5) Young, Female Smile. PSEM paths always end on superpixels relevant to the face.

a reasonable comparison with PSEM and IR (CEM-PP generally finds longer explanations that are harder to decipher). We append a (+) or (-) to words selected by LIME to indicate positive or negative relevance. Table 3 shows two examples of documents where these methods are illustrated.

Document 1 shows a correctly classified document where PSEM finds a slightly sparser explanation than IR. Both methods offer intuitive explanations, but the greedy path sometimes removes words early that result in not being able to find the (potentially) sparsest sufficient explanation. LIME gives sensible words, although the positive vs. negative relevancies are counterintuitive here.

Document 2 shows a misclassified document where IR selects no words, meaning an ordering of the words exists where all words can be removed one by one without changing the rec.autos classification. Indeed, an empty document is classified as rec.autos by the trained model, and since all inputs result in some classification, such results should not immediately result in designating the classifier as untrustworthy. PSEM selects words 'bought', 'new', 'used' which could easily be in a document about automobiles (while we recognize these words do not have to be about buying new/used cars, no category except for rec.motorcycles is about commodities that are bought new/used). LIME offers a similar story with the words 'bought' and 'tire', but again, it is not clear how big LIME's explanation should be, whereas PSEM's length is determined by definition. This is a good example of how PSEM can be used to locate potential issues with a classifier - even though other categories can discuss commodities being sold, it appears that there is a bias of the classifier towards predicting such samples to be about automobiles.

## 5.2 Color Image data: CelebA

We next experiment on color images where the PSEM path is over superpixels rather than individual features (by solving Problem (3) as discussed in Section 3). The CelebA (Liu et al., 2015) dataset contains images of celebrity faces along with 40 features describing things such as hair color, face shape, etc. We follow the setup of Luss et al. (2021) and explain an 8-class (by joining 3 binary classes) classifier, a Resnet50 model (He et al., 2016), that predicts the following three classes: sex (male/female), smiling (yes/no), and age (young/old). While, we apply PSEM here to Luss et al. (2021), we acknowledge that there are various ways to handle the masking of pixels, e.g., Fong & Vedaldi (2017) and Fong et al. (2019), but analyzing different stability penalties is outside the scope of this paper, where the goal is to derive paths of sufficient explanations which could be done for these works as well.

The results on ten images are shown in Figure 2. CEM-PP uses the same sparsity parameter as PSEM-4 which is how CEM-PP would generally be used (to find a minimally sufficient explanation). The number of

superpixels in LIME is set to the same used in CEM-PP. All PSEM explanations are smooth transitions to the sparsest explanation given by PSEM-4. This is a key advantage over CEM-PP regarding trust in the explanation. On the left, in rows 1 and 2, PSEM-4 and CEM-PP are both very sparse and uninformative, so a user might not trust that the classifier is focused on these areas (as noted earlier, the classifier must predict something for these images thus resulting in "garbage in, garbage out" results from classifying such sparse images). There is no guarantee that a stable sufficient path to the original image exists from CEM-PP (meaning the classifier is not focused on CEM-PP when classifying the original image), but if a user wanted a denser explanation, it is guaranteed along the PSEM path. In rows 2 and 3 on the left, CEM-PP does not highlight the mouth, while PSEM-4 does and with similar sparsity, illustrating that PSEM often leads to more realistic explanations. The last row on the left shows an example where all methods find what appear to be sufficient and relevant features. PSEM identifies the forehead versus the cheek identified by CEM-PP; the forehead is likely a better predictor of age. We also show LIME results, which mostly lack interpretability. This is not necessarily a fault with LIME explanations, as LIME explains using relevancy rather than sufficiency. Note that LIME is set to have the same number of features as CEM-PP for fair comparisons. Similar observations can also be made in examples on the right-hand side of Figure 2.

### 5.3 Image data: MNIST Handwritten Digits

The MNIST dataset is comprised of handwritten digit images 0-9. We trained a convolutional neural network from Dhurandhar et al. (2018) which uses two blocks, each composed of two convolutions followed by a pooling layer, followed by three fully-connected layers. Note that Dhurandhar et al. (2018) also include an additional regularization of the form $\|\boldsymbol{\delta} - \mathrm{AE}(\boldsymbol{\delta})\|_2^2$, where $\mathrm{AE}(\cdot)$ is an autoencoder; this term ensures that the learned PP lies near the true manifold, however Dhurandhar et al. (2018) only uses it on MNIST experiments where a sufficient autoencoder is learned. We similarly use an autoencoder only for MNIST experiments and thus left it out of the formulations for clarity. The same autoencoder architecture from Dhurandhar et al. (2018) is used to keep PSEM explanations near the image manifold.

Results are illustrated in Figure 3. PSEM, CEM-PP, and LIME are compared across a single image each from 0-9. PSEM outputs a path of five images where PSEM-$i$ labels the $i^{\mathrm{th}}$ image along the path where the sufficient number of pixels is decreasing from PSEM-1 to PSEM-5. For CEM-PP, the sparsity penalty in CEM-PP is set to the same sparsity penalty used in PSEM-3 in order to give a comparison with CEM-PP that is not too sparse (sparsity levels differ because PSEM and CEM-PP have different regularizations).

A key observation is that stability of the PSEM path results in sufficient explanations that clearly distinguish each of the digits. Better stability of PSEM is seen on the digit 1 where CEM-PP fails to find a minimally sufficient set. The different regularization of PSEM allows for better control to learn an intuitive path of explanations, where the sparser explanations even focus on different regions of the digit. CEM-PP has an intuitive explanation for the 0 (but very sparse for 2-7 and 9), and as already noted, if a user wanted a denser explanation, there is no guarantee that adding pixels could remain in the same class, and simply decreasing the sparsity regularization could lead to a completely different explanation which would be counterintuitive.

PSEM offers a smooth path of explanations to the user. In particular, we see that PSEM converges to a notably different explanation than CEM-PP for 2, 4, 5, 6, 7, and 8. PSEM shows that the tail is distinguishing for the 2, the $\cup$ shape for the 4, a full loop for the 6, the corner of the 7, and the $\mathbf{x}$ part of the 8, which is clearly unique to that digit. LIME mostly finds the entire digit positively relevant for the prediction, which could build trust in the classifier, but does not add additional useful information into how it might be making decisions. This does not mean that LIME isn't useful; indeed, LIME could be used to analyze why it was not predicted something else since it allows for targeted explanations, unlike PSEM or CEM-PP.

## 6 User Study

We conducted a user study to investigate the information derived from PSEM in comparison with other local explainability methods: CEM-PP and LIME. Our setup is in similar spirit to user studies conducted in Ribeiro et al. (2016) and Singh et al. (2019); for each explanation method, users were provided with two explanations, one for each of two different models, and had to select which model was more accurate.

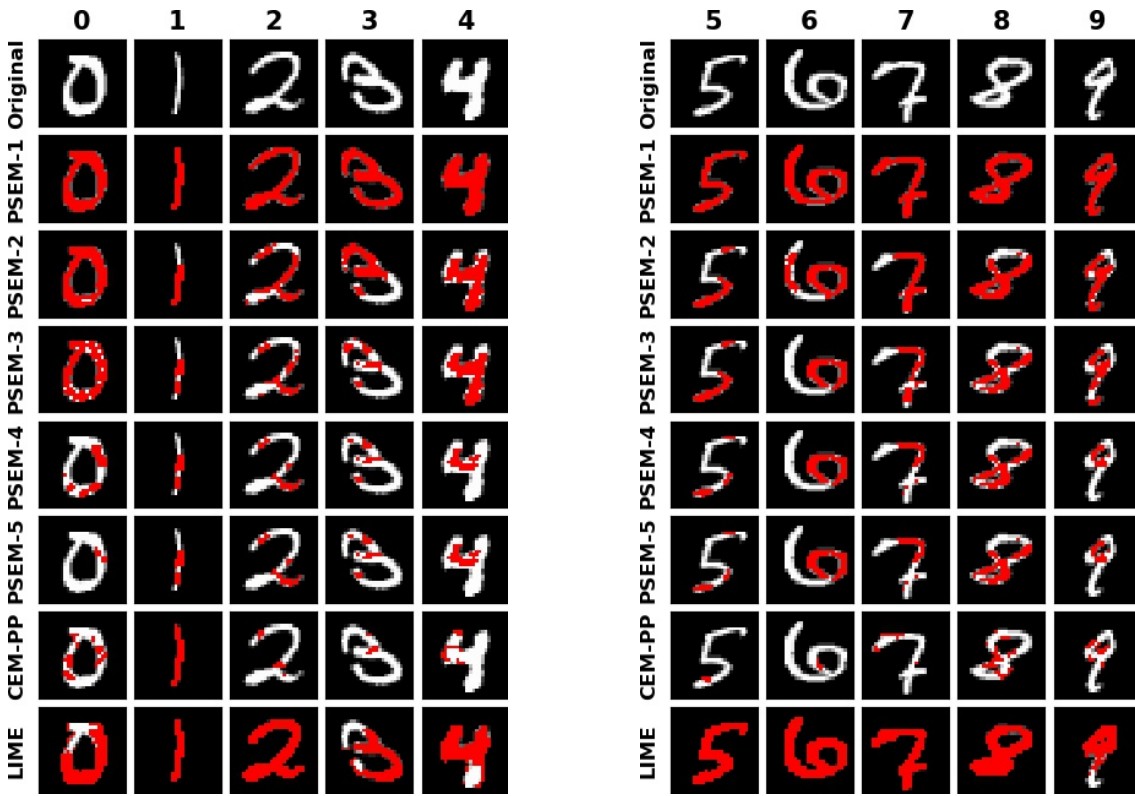

**Figure 3:** PSEM, CEM-PP, and LIME are demonstrated on digits 0-9 from the MNIST dataset. Red pixels highlight what is sufficient for PSEM along the path, what is sufficient for CEM-PP using the same $\beta$ as in PSEM-3, and what is deemed positively relevant by LIME.

Our user study is done on the full 10-class MNIST dataset. We determined MNIST to be the most appropriate choice; text samples are long and would make the study laborious, the loan example requires domain expertise, and color images use an extended model that is not as widely applicable as the first model via (2). Furthermore, while MNIST might appear simplistic, the task for participants described below is not easy, as witnessed by the accuracies in Figure 5. In order to keep survey complexity manageable, we compare with the most popular XAI tool LIME and most relevant CEM-PP.

We have two models: a 99.9% accurate CNN from Dhurandhar et al. (2018) which uses two blocks, each composed of two convolutions followed by a pooling layer, followed by three fully-connected layers and a 60.2% accurate NN with two hidden layers. A digit is displayed along with one explanation per model. For each digit, we display the same explanation method (PSEM, CEM-PP, or LIME) applied to both the accurate CNN and the inaccurate NN, randomly assigned as Model A or B. The question for the example in Figure 4 reads "The two predictions are 6 and 5. Which model below predicted a 6?" and the user must select one of three options: Model A, Model B, or "Cannot tell". A user might realize that the remaining highlighted piece in Model A's explanation would not appear in a handwritten 5 so they select Model A (and they would be correct in this case).

The survey contains explanations for ten individual digits with three questions per digit (one per explanation method) for a total of thirty questions. In order to make the study fair across methods, the instructions give brief descriptions for the three explanation types. Each question tells the user which of the three explanation types is given and the user does ten sets of three questions (one per explanation type) on the same digit. We deem this fair because any user of the explanations would know what kind of explanation tool they are using. It would be a biased study if a user had to answer a question for a CEM-PP explanation thinking this explanation highlighted the positively relevant features as LIME does. Recall that the goal of the study is to investigate whether a user can use an explanation method to distinguish between two different models.

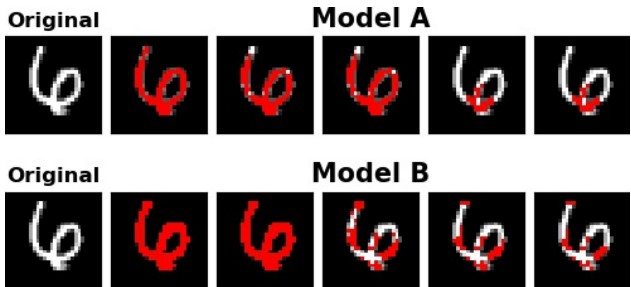

**Figure 4:** User study example with two models. Models A and B predicted 6 and 5. Participants are shown an explanation (here, PSEM) for each model's predictions and must select which model predicted 6. Other questions do the same with CEM-PP and LIME.

Knowing the type of explanation does not bias this study; it is a fact that each method offers different information. This is a key difference from the study in Ribeiro et al. (2016) where both explanation methods output relevant features. The order of explanation methods is randomized for each digit. We used the Google Forms platform to execute the study. It was sent to ≈ 50 participants of which 37 responded, with backgrounds in various quantitative disciplines. We chose this demographic as it was shown that people with such backgrounds (engineers, scientists, etc.) are the main consumers of such automated explanations (Bhatt et al., 2020). The User Study gave participants 10 sets of 3 consecutive questions pertaining to CEM-PP, LIME, and PSEM. Screenshots can be found in the Appendix.

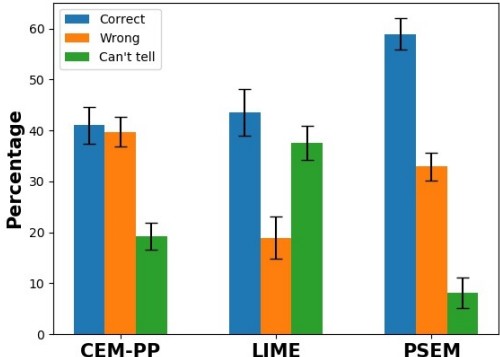

| Preference | Expl. Type | % Correct |
|---|---|---|
| | CEM-PP | 41.5 ($\pm$7.0) |
| CEM-PP | LIME | 43.8($\pm$5.5) |
| | PSEM | 60.0($\pm$3.4) |
| | CEM-PP | 32.2 ($\pm$9.4) |
| LIME | LIME | 47.8 ($\pm$7.8) |
| | PSEM | 42.2 ($\pm$5.2) |
| | CEM-PP | 46.0 ($\pm$7.5) |
| PSEM | LIME | 40.7 ($\pm$3.6) |
| | PSEM | **68.0** ($\pm$6.7) |

**Figure 5 & Table 4:** Left: Accuracy of user study participants at distinguishing between two models based on PSEM, CEM-PP, and LIME. PSEM offers extra information that helps users better discriminate between predictions in a manner better than the other explanations. Right: Results to exit question asking which explanation the user found "most useful for explaining why each model made their predictions." Participants that preferred PSEM performed particularly well on PSEM questions; i.e., extra information is more meaningful to those that prefer more information.

Statistically significant results are displayed in Figure 5. Across three methods, CEM-PP, LIME, and PSEM, we see the percentage of correct, wrong, and cannot tell with error bars showing one standard error. The percentage correct for PSEM is 58.9%, 43.5% for LIME, and 41.1% for CEM-PP, with pairwise t-test p-values of 0.002 and 0.004 comparing PSEM to LIME and CEM-PP, respectively (i.e., benefit of PSEM is statistically significant). This does not tell the complete picture as there are a significant number of "cannot tell", mostly for LIME, so those % correct numbers could have been different if participants were forced to choose; however, in practical applications, an explanation where the user cannot discern information is on par with a bad explanation. These results show that PSEM offers users useful information that they can disseminate. Whether users are right or wrong, it is interesting to note that users are much more comfortable making decisions based on the PSEM path.

A final exit question asked which explanation type the user found "most useful for explaining why each model made their predictions," and the results were 40.5%, 35.1%, and 24.3% for PSEM, CEM-PP, and LIME, respectively. While one can debate whether this question is biased because PSEM offers more information,

it is worth noting that participants that preferred PSEM did much better on PSEM questions than those that preferred LIME or CEM-PP. By grouping participants according to explanation preference, Table 4 shows that only those that preferred PSEM performed particularly well, and furthermore only on PSEM questions. Those that preferred LIME performed best on LIME (however still not good performance). Those preferring CEM-PP did not perform best on any explanation type, while those preferring PSEM also performed best on CEM-PP. This implies that the extra information of PSEM does not overwhelm but rather significantly improves understanding. A deeper look at the data also shows interesting cases such as preference for CEM-PP where the user scored 8/10 on PSEM and 6/10 on CEM-PP, and noted that he/she generally moved to the last image of the PSEM explanation. So either subconciously the path helped this user or the PSEM path actually did converge to better explanations than CEM-PP (at least in the view of this particular user).

## 7    Discussion

PSEM is more principled than individually learning pertinent positives at different sparsity levels or removing least important features as in input reduction because the stability of the path removes those sufficient solutions that are likely random artifacts of the classifier. Crucially, such stability, along with the comprehensibility it creates, is required to build trust in the explanation and to make sufficiency-based explanations more widely used. Consider the final row in Figure 2, where the difference between PSEM-4 and CEM-PP is that one likely uses the forehead to classify age while the other likely uses the cheek. Clearly, the forehead is more useful for this task. The cheek was removed by PSEM-3 along the path because a sufficient set of features did not not require the cheek, but there is no guarantee that removal of the forehead instead would have been sufficient, i.e., the CEM-PP result with the cheek is possibly a classifier artifact, similar to how the classifier must predict something on a blank image.

We have also shown that PSEM offers more realistic scenarios as seen in the Introduction; CEM-PP offers a point explanation and it is not clear if those values are lower bounds to maintain the class, while the path in PSEM provides much better intuition regarding this aspect, in addition to presenting users with more realistic possibilities of possible actions.

Interest future directions are inspired by Ignatiev et al. (2021) and Darwiche & Ji (2022), where they seek to learn multiple sufficient explanations for an individual sample. In the case of logical formulae, it is clear how to define the length of explanations, but it is not apparent for a continuous set of features, as with the loan applicant example in the Introduction. However, we can design algorithms for learning multiple paths, e.g., by starting PSEM from different initial points or by using different $\beta$ parameters to derive different paths. Another direction for future research is to apply PSEM to explanations beyond pertinent positives, e.g., semi-factual explanations. In the our case, the monotonicity of the pertinent positive explanations is crucial to the path because it is clear how to formulate the monotonic removal of features. As such, PSEM is a meta-approach, where the base explanation could be pertinent positives or semi-factuals. However, in the latter case, it is not obvious how to formulate a general (smooth) removal of features, and remains an interesting future direction.

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

## A  PSEM Implementation and Reproducibility

As previously noted, PSEM contains several parameters that need to be tuned. While this process can be extensive, it is important to consider that, for a given dataset and application, the parameters need only be tuned once, after which, an arbitrary number of predictions can be explained. Table 5 lists the hyperparameters used to get the results throughout the paper. Note that parameter $c$ was initialized to 10 for each experiment and the PSEM code searches over that parameter if a minimally sufficient solution cannot be found for that value of $c$. Regarding the importance of each parameter, sparsity parameters $\beta$ were the last parameters to be tuned. First, for a relatively small $\beta$, we found it best to focus on tuning $\eta$ and $\kappa$. $\eta$ was the least sensitive parameter, and $\kappa$ was relatively easy to tune, once the user gets an idea of what value of $\kappa$ (i.e., confidence) was too small to find any PPs. Most sensitivity in tuning was found to be due to $\beta$ which was done with other parameters already fixed.

The PSEM implementation adapts CEM-PP code from `https://github.com/IBM/AIX360`. PSEM entails sequential calls to modified versions of CEM-PP. Modifications include adding regularization terms $\eta\|\delta - \delta_{i-1}^*\|_2^2$ and $\eta\|M_{\boldsymbol{\delta}} - M_{\boldsymbol{\delta}_{i-1}^*}\|_2^2$ for problems (2) and (3) and projections to maintain consistency along respective

**Table 5:** Parameters used for various experiments

| Dataset | $\beta$ | $\eta$ | $N$ | $\kappa$ |
|---|---|---|---|---|
| MNIST | {0.0001, 0.001, 0.01, 0.1, 1.0} | 10.0 | 5 | 0.75 |
| HELOC | {0.00001, 0.0001, 0.001, 0.01, 0.1} | 30.0 | 5 | 0.2 |
| CelebA | {0.001, 0.005, 0.01, 0.05} | 0.01 | 4 | 0.02 |
| 20 Newsgroups | {0.0001, 0.0005, 0.001, 0.005, .1} | 50.0 | 5 | 0.5 |

**Table 6:** Results comparing stability and fidelity of path methods on 20 Newsgroups. Stability measures the % of features at each index ($> 1$) along a path that also appeared at the previous index. Fidelity measures the % of predictions, using features at each index, that are equal to the original prediction. Metrics are averaged over 64 samples with one standard error shown in parentheses.

| | Expl. Type | Index Along the Path | | | |
|---|---|---|---|---|---|
| | | **1** | **2** | **3** | **4** |
| # Feat. | ALIME | 26.5 (0.8) | 9.9 (0.4) | 4.4 (0.3) | 2.3 (0.2) |
| | LIME | 36.3 (1.0) | 14.2 (0.7) | 6.7 (0.4) | 3.7 (0.3) |
| | CEM-PP | 44.6 (0.9) | 3.6 (0.3) | 6.0 (0.3) | 6.8 (0.4) |
| | PSEM | 44.6 (0.9) | 34.0 (0.9) | 13.2 (0.4) | 7.3 (0.3) |
| Stability | ALIME | – | 99.9 (0.1) | 99.8 (0.2) | 100 (0.0) |
| | LIME | – | 99.6 (0.3) | 99.3 (0.3) | 99.7 (0.3) |
| | CEM-PP | – | 100 (0.0) | 48.7 (3.1) | 64.8 (3.4) |
| | PSEM | – | 100 (0.0) | 100 (0.0) | 100 (0.0) |
| Fidelity | ALIME | 81.8 (4.7) | 63.6 (5.9) | 36.4 (5.8) | 18.2 (4.7) |
| | LIME | 95.5 (2.6) | 95.5 (2.6) | 86.4 (4.2) | 63.6 (5.9) |
| | CEM | 100 (0.0) | 100 (0.0) | 100 (0.0) | 100 (0.0) |
| | PSEM | 100 (0.0) | 100 (0.0) | 100 (0.0) | 100 (0.0) |

paths. The $l_2$ regularizations from CEM-PP are removed. One implementation caveat for explaining large models is to create a single object, referred to as AEADEN in CEM-PP code, where parameters that vary along the PSEM path $(\beta_i, \delta^*_{i-1}, M_{\delta^*_{i-1}})$ are implemented with placeholders. This object can then be used sequentially to solve the PSEM subproblems by passing appropriate parameters (and solutions of previous iterates) at each iteration to the `attack` function of CEM-PP. If one rather creates a new AEADEN object at each iteration and calls the respective `attack` functions, the iterative creation of new AEADEN objects, which replicate the classification model, can quickly use up memory resources. We implemented IR. All experiments used 1 GPU and up to 16 GB RAM.

## B    Additional Quantitative Results

See Table 6 for quantitative metrics on the 20 Newsgroups dataset, where improved stability of ALIME comes at the cost of fidelity rather than interpretability, i.e. improved stability with fewer features but reduced fidelity for (A)LIME.

## C    Additional comments about User Study

Figure 6 details the instructions for participants to read before taking the user study. Figures 7 and 8 show two examples of such sets of questions, for a digit 3 and digit 6, respectively. The correct answers for the digit 3 questions (corresponding to CEM-PP, LIME, and PSEM) are Model A, Model B, Model A. Note that for CEM-PP and PSEM, the inaccurate model cannot find minimally sufficient solutions, and LIME typically highlights more positively relevant pixels for the accurate model. The correct answers for the digit 6 questions (corresponding to CEM-PP, LIME, and PSEM) are randomly also Model A, Model B, Model A. CEM-PP and PSEM both highlight upper parts of the left part of the loop in the 6, which would not be present in a 5. The PSEM path additionally highlights other parts that could help distinguish the 6 from other digits, such as an 8 which has a lower loop like a 6. Again, LIME highlights more positively relevant

pixels for the accurate model, which implies that LIME is giving the information that it is designed to give, while being of different use than minimally sufficient descriptions.

**Figure 6:** User Study Instructions.

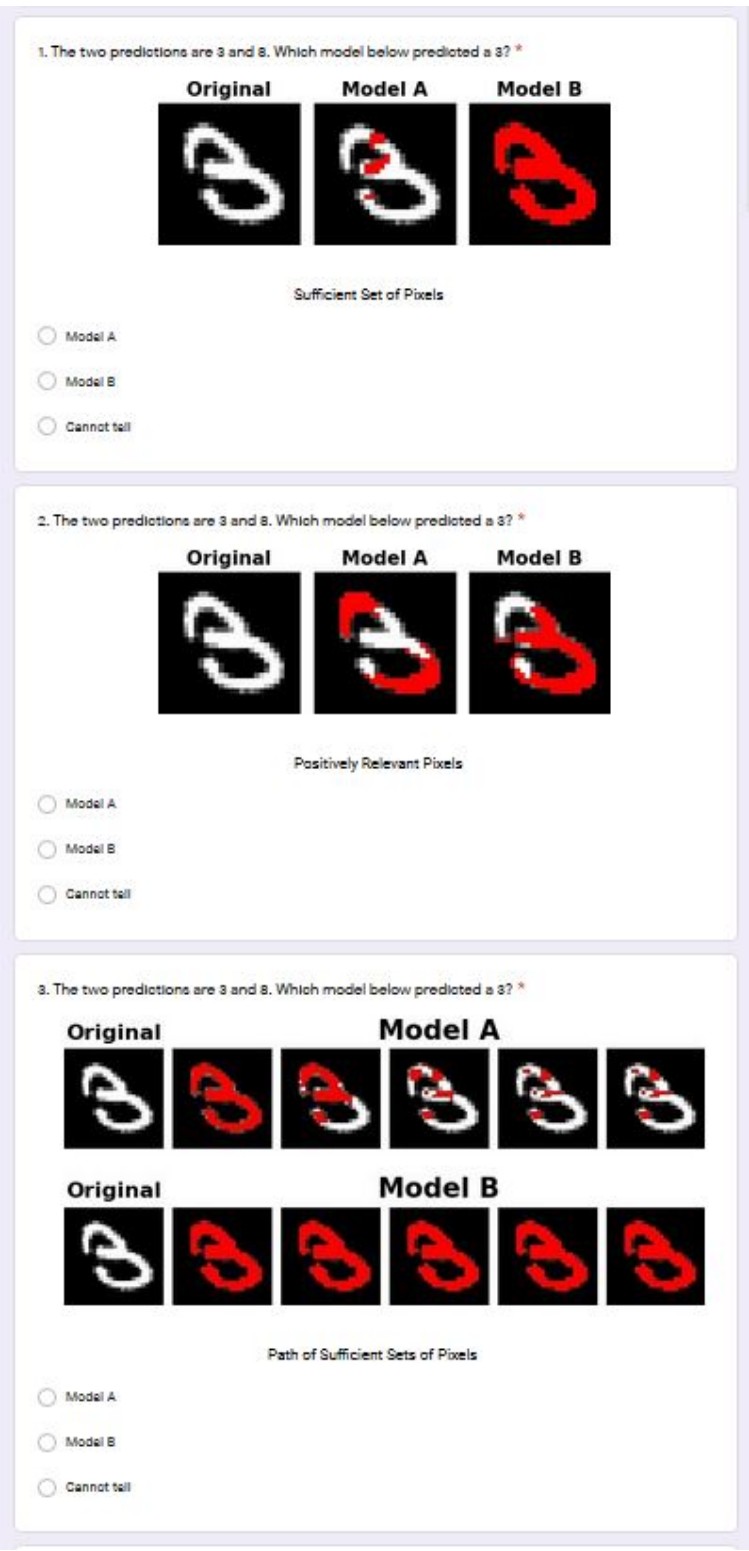

**Figure 7:** User Study example on the digit 3.

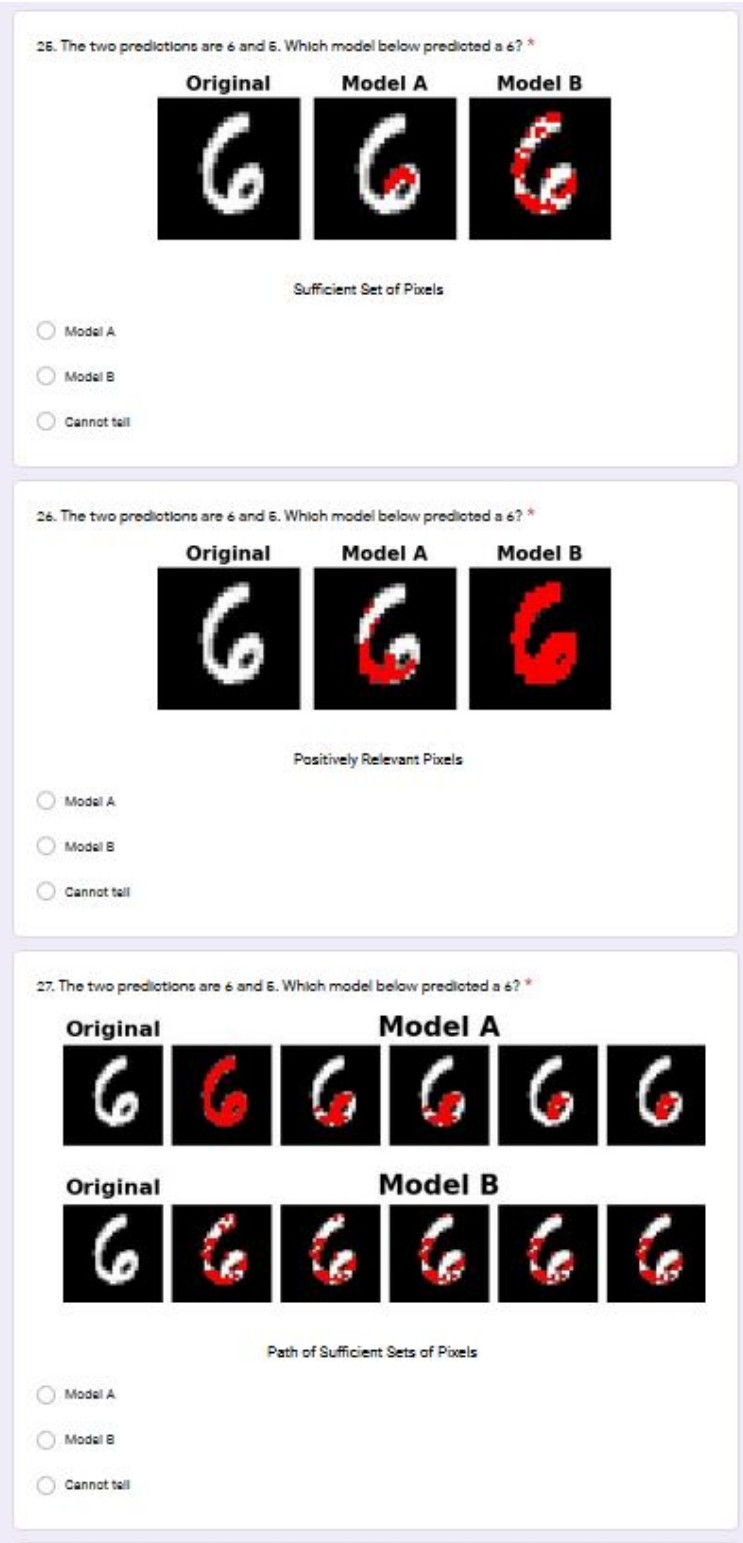

**Figure 8:** User Study example on the digit 6.

