# OpenReview forum: "When Stability meets Sufficiency: Informative Explanations that do not Overwhelm"
_TMLR — Accepted by TMLR_

### Review · Reviewer_Evvb · 2024-07-16

**Summary Of Contributions:**

The paper considers the problem of interpreting black-box machine learning models, and it proposes an algorithm to generate a sequence of modified inputs that maintain the prediction from the original input while becoming increasingly sparse. When determining the sequence, the method aims to maintain "stability" (each modification is not too different from the previous one), "sufficiency" (each modification results in the same prediction), strictly decreasing size (features altered in one modification must remain altered in the next one), and "monotonicity" (features are modified in a single direction). These criteria are represented in an optimization problem, shown in eq. 1, that the authors propose solving iteratively using proximal gradient descent (FISTA).

The method is evaluated through a couple quantitative metrics, 2/3 of which correspond to criteria in the eq. 1 optimization problem. The authors then show qualitative examples for CelebA classification, Newsgroups classification, and MNIST digit classification. Finally, the authors perform a user study using MNIST: they show users explanations for two models that made different predictions, and test whether users can identify which model is which.

**Audience:**

No

**Broader Impact Concerns:**

None.

**Claims And Evidence:**

No

**Requested Changes:**

My largest concern with this paper is that it ignores a long line of highly relevant existing work. Many of the challenges recognized in those papers are ignored here, and none of those methods are used in experimental comparisons. I would request that this paper be rewritten with a better framing of how it fits into the literature, and a more careful evaluation of what is new and valuable here.

**Strengths And Weaknesses:**

### Strengths

The proposed method is applied to multiple data modalities.

### Weaknesses

The basic idea of this paper is to find input perturbations that adjust the prediction in a desired way (e.g., maintaining it) and with controlled levels of sparsity. There are many works on this topic that aren't cited here, including but not limited to: "Learning to explain: An information-theoretic perspective on model interpretation", "INVASE: Instance-wise variable selection using neural networks", "Have we learned to explain? How interpretability methods can learn to encode predictions in their interpretations", "Interpretable explanations of black boxes by meaningful perturbation", "Understanding deep networks via extremal perturbations and smooth masks", "Real time image saliency for black box classifiers", "Object detectors emerge in deep scene CNNs", "Explaining image classifiers by counterfactual generation". A couple questions that typically arise in these works, which ideally would have been discussed here, are:

- How do exactly you want to modify features? Do you want to perturb them to specific counterfactual values (like we do in counterfactual/recourse explanations), or do you want to basically mask/remove the feature's information? In this paper, I'm not sure what the authors are going for. On the one hand, it seems like they don't want to perturb to specific values because they parameterize the perturbations as $\delta \in [0, 1]^n$ and say they're optimizing over relaxed binary variables; indeed, for most of the image experiments, it looks like they have each $\delta$ entry equal to either 0 or 1. On the other hand, if they actually wanted to optimize over binary variables, there are better ways to do so (as shown in these works, e.g., optimizing using the Concrete/Gumbel reparameterization trick), and the authors at times highlight the monotonic nature of feature changes (e.g., in Table 1). Overall, this aspect of the method seems a bit confused and imprecise.

- How do you avoid adversarial perturbations? When optimizing over the input to deep learning models, it is easy to achieve arbitrary predictions in any neighborhood of the input space, but the chosen input may be incorrectly classified due to quirks in the nonlinear prediction function. Finding such inputs in your optimization problem would be misleading to users. This issue is less problematic when optimizing over truly binary perturbations (e.g., with the Concrete distribution), or when incorporating strong regularizers (see Fong et al., 2017), but your optimization approach with real-valued $\delta$ makes adversarial perturbations easy to find. Interestingly, a few of the experiments seem to sidestep this risk using small models, which may reduce the risk of adversarial perturbations. But overall, this risk is not recognized or mitigated by the authors.

- When optimizing over the input perturbation, do you want to maximize or minimize the true class? Both approaches have been explored extensively, and sometimes simultaneously (see "Real time image saliency for black box classifiers"). I don't think either approach is clearly right or wrong, but some rationale would be useful - in this work the choice seems arbitrary.

In comparison to those previous works, finding a sequence of modifications seems different and new. But it's a pretty trivial extension, because all of those algorithms can be run with different levels of sparsity. Connecting the sequence of solutions via the "stability" constraint/regularization is a small contribution, but also straightforward.

Some other concerns:

- In the introduction, the authors mention certain explanations being difficult to act on because of high sparsity. If you want an explanation to be actionable, reducing the sparsity, or providing a sequence of explanations with different sparsity is a pretty blunt solution. You should probably use a counterfactual explanation that optimizes for specific changes in the input space, with a reliable approach to avoid adversarial solutions, and constrain the optimization to only affect actionable features. It's not a good justification for this work.

- In the introduction, the authors mention that constraining each input perturbation in the sequence to be similar to the previous one may help keep inputs on the data manifold: "Our measure of stability [...] connects each solution along the path, in effect forcing the applicant profiles to remain realistic." That's not true and is another poor argument in favor of this method. You could of course create a sequence of perturbations that become progressively less realistic and drift into parts of the input space with no support under the data distribution.

- The proposed method has a number of hyperparameters that can strongly affect the results. The discussion of how these are tuned is deferred to the appendix, where the authors describe the difficulty of tuning each value but not the criteria for making their final decisions. I suspect, but it would be nice to have confirmation, that decisions were made manually based on which results looked good to the authors?

- In the eq. 1 optimization problem, the objective term $f_\kappa$ seems redundant which the constraint in (c). Have the authors thought about to what extent each of these is necessary? Ideally there would be an ablation to justify the choice to keep both.

- When discussing the use of their approach for color images, the authors explain that RGB and grayscale images require a different treatment: "color images require a new perspective because, while adding or removing features from grayscale images simply reduces to increasing and decreasing pixel intensities, we cannot similarly optimize over pixel intensity in RGB space as there is no concept of background and the results would be unrealistic. Rather, the concept of feature removal is done by masking superpixels (segments composed of multiple pixels) that result from an image segmentation." I certainly agree that your masking approach risks producing unrealistic inputs, but the presence of color or lack of background does not clearly necessitate the use of superpixels. When you perturb color images, you end up setting regions to gray, which seems to function just fine as background, right? So why not do this for individual pixels? Is it possible that superpixels serve another purpose that the authors aren't acknowledging here, such as mitigating adversarial or discontinuous solutions? If so, it would be best to explain that, and discuss why you don't explore other approaches, for example the total variation penalty adopted in Fong et al., 2017.

- The section 4 experiments are nearly a sanity check, because 2/3 criteria are built into the proposed method.

- Why does the Newsgroups experiment use such a simple model? I get the argument that you can test explanation reliability for any model, even a very simple one, but it's concerning that the chosen model is so far from state-of-the-art. Why, would any difficulties arise with using a modern approach, e.g., fine-tuning a BERT model for text classification?

- The qualitative results for Newsgroups, CelebA and MNIST are difficult to assess. I would be interested to know if other reviewers have a stronger feeling that this demonstrates usefulness, but to me it does not.

- The user study is perhaps the most compelling evaluation approach in the paper. There are some issues, for example I would curious to know exactly how the authors described each method to users (whether the descriptions were simple enough to understand and unbiased), and how the authors could mitigate the unfairness caused by their method simply providing more information than LIME. But it's neat that users could more reliably identify which model was which.

- Since the proposed method is, I believe, basically aimed at removing information rather than perturbing features to specific values, LIME is a strange choice of comparison. LIME is neither optimizing over which features to remove nor perturbing to specific values: it effectively tests removing all possible subsets of features and reports the average impact of removal by solving a weighted least squares problem. Many of these methods are linked under the framework of removal-based explanations (see "Explaining by Removing: A Unified Framework for Model Explanation"), but it's nonetheless a strange comparison. Getting back to the long list of methods above, I believe a more informative comparison would be with respect to those. By ablating specific design choices, we could get a clearer sense of which ideas here are new and useful: for example, the way the perturbation is parameterized, the optimization algorithm, the regularization / mitigation for adversarial examples (none are used here), whether we aim to minimize or maximize the true class's prediction, how removed features are represented, etc.

---

> ### Author Response · Authors · 2024-09-25
> **Response to your review [1/4]**
>
> Thank you very much for the detailed review and taking the time to go through our paper. We believe there are several misunderstandings with this work regarding the literature on perturbations and your questions that arise from them that we hope to clear up. We follow these responses by addressing your other concerns.
>
> > Regarding "There are many works on this topic that aren't cited here, including but not limited to..."
>
> We address posthoc local explanations for machine learning models. This means that we are given a model to explain along with an instance for which we wish to explain why the model gave its particular prediction on this instance. Sufficient explanations are specifically a type of attribution method and we rightly compare with other attribution methods. As noted in the Introduction in Footnote 1 (and through the example), counterfactual explanations offer complimentary information; one should strive to get both types of explanations but we should not compare one versus the other. This is why Dhurandhar et. al. (2018) derive both CEM-PP along with counterfactuals in the same work. The same can be said of hierarchical explanations. We further discussed explanations that offer different types of information in the second paragraph of Section 2 on Related Work.
>
> Specifically regarding the list of papers suggested, we note that the first three papers ("Learning to Explain: An information-theoretic persepctive on model interpretation" which offers a method called L2X that trains a neural network explainer, "INVASE: Instance-wise variable selection using neural networks" that trains a predictor, baseline, and selector neural networks to offer explanations from the selector, and "Have we learned to explain? How interpretability methods can learn to encode predictions in their interpretations" that offers a method called REAL-X that also trains a global selector network to identify important features) all require training data to train different neural networks that are used to do feature attribution. This is a large departure from our scenario which does not assume training data (we do assume a small validation set for tuning hyperparameters but this is very different from requiring a training set to train neural networks). We also note that the third paper specifically offers the REAL-X method in order to address the issues of the first two methods L2x and INVASE. As the third paper notes, these are all joint amortized explanations (JAMS) that require training a network as in equation (3) of the REAL-X paper.
>
> The two papers "Real time image saliency for black box classifiers" and "Object Detectors Emerge in Deep Scene CNNs" are specifically about image classifier explanations, and as discussed in the paper, we focus on explanation methods that are general to various modalities. Additionally, as with the above papers, "Real time image saliency for black box classifiers" requires training a large neural network for inferring image masks and assume 250,000 training images, again outside of our assumptions. The paper, "Object Detectors Emerge in Deep Scene CNNs", is tailored to explain Convolutional Neural Networks, and particularly focus on the pooling operations, which again is not a general explanation method.

---

> > ### Author Response · Authors · 2024-09-25
> > **Response to your review [2/4]**
> >
> > The remaining three papers discussed are also solely applied to images (they could be applied to tabular/text although it might not make sense given the infilling used). We agree these papers are related to Luss et.al. (2021) cited in the paper, however, please note that we did also cite the most recent of those works, "Understanding Deep Networks via Extremal Perturbations and Smooth Masks" by Fong et.al. (2019) and noted in the related work that this paper, along with Luss et.al. (2021) are extended to our framework. In practice, we implement a version of Luss et.al. (2021) that removes a regularization term they included for maintaining latent attributes in an image resulting in our equation (3) (we add the stability regularization). Luss et.al. (2021) and Fong et.al. (2019) are otherwise different ideas for the same problem - Fong et.al. (2019) applies smooth masks over individual pixels while Luss et.al. (2021) considers smooth masks according to segmenting the image first. Luss et.al. (2021) could incorporate blurring or other infilling for the removed parts of the image but such an investigation is outside of the scope of this paper. The other two papers "Explaining Image Classifiers by Counterfactual Generation" and "Interpretable explanations of black boxes by meaningful perturbation" are very similar papers - see Figure 5 of Chang et.al (2019) for a comparison - and note that Fong et.al. (2019) builds on the Fong and Vedaldi (2017) paper. The main differences in these works is how infilling is done to create masks/perturbations. We have cited all of the above works, with appropriate contexts and clarifications, in the related work section 2 of the revision, but please consider that these questions about perturbations are outside the scope of this work. The same questions are asked in other explanation methods such as LIME; it is well-known that the local simulator has a great impact on the quality of the explanations.
> >
> > > How do exactly you want to modify features?
> >
> > We parameterize perturbed examples according to the the space of the input. Please refer to the first paragraph of Section 3.  For the different modalities considered, we assume different input spaces, and then constrain $\delta$ to lie in these spaces. We believe there is another misunderstanding here. Counterfactuals specifically seek a minimal change in features that changes a prediction, while a Sufficient Explanation seeks the minimal features that remain in order to maintain the original prediction. This paper solely deals with the latter. In tabular data, features are not necessarily removed/masked but rather reduced. We differ between these explanations and why they are complimentary in the first paragraph of the Introduction (along with Footnote 1). Also note that we only use the relaxed binary variables for the color images.
> >
> > > How do you avoid adversarial perturbations?
> >
> > Again, we searching for perturbations that maintain a classifier's original prediction, not for adversarial perturbations that change the classification. Note that the "Explaining Image Classifiers by Counterfactual Generation" you cite differs between these problems SDR (as the adversarial problem) and SSR (which is what we term the pertinent positive or minimal sufficiency problem). The risk for sufficient solutions is that of discovering what we call irrelevant artifacts of an image. We discuss in Section 4 how stability seems to reduce the risk of converging to an irrelevant artifact along the path. We further discuss this aspect in the conclusion regarding how CEM-PP might deem a cheek as relevant for predicting age rather than foreheads that are discovered by the path algorithm. We also point out that the best way to reduce the risk of adversarial pertubations in a neighborhood of a sample would be to include regularizer based on a generator that maintains the perturbations lying in the manifold of the input space. This was done with an autoencoder for MNIST in Dhurandhar et.al. (2018), but applying this in general applications is difficult and outside the scope of this paper.

---

> > > ### Author Response · Authors · 2024-09-25
> > > **Response to your review [3/4]**
> > >
> > > > When optimizing over the input perturbation, do you want to maximize or minimize the true class?
> > >
> > > As with the "Explaining Image Classifiers by Counterfactual Generation" paper, "Real time image saliency for black box classifiers" does look at both the adversarial perturbation and minimal sufficiency problem, and as we describe above, we focus on the latter. As we stress in the Introduction, these are two different problems, and each explanation offers different information. For example, if you are the loan applicant in the Introduction, which explanation you'd be interested in depends on your situation. We are very specific in this paper that we focus on the minimal sufficiency problem (also called the Smallest sufficient region or the Smallest Supporting region in the other literature). In the text, we word this as having our loss penalize perturbation that differ from the original classification.
> > >
> > > > Regarding "You should probably use a counterfactual explanation that optimizes for specific changes in the input space ... It's not a good justification for this work."
> > >
> > > We respectfully disagree with this assessment. Counterfactual explanations and sufficient explanations are two complimentary types of explanations that answer different questions and we specifically point this out in the Introduction. Sufficient explanations are a type of attribution method, and hence our comparisons with LIME, CEM-PP, and Input Reduction. It is very valid to ask what reduction in a feature space would maintain a prediction and we give a very compelling and realistic motivation for such in the Introduction with the loan applicant example. Adversarial explanations do not always answer the question of interest to a user.
> > >
> > > > Regarding "You could of course create a sequence of perturbations that become progressively less realistic and drift into parts of the input space with no support under the data distribution"
> > >
> > > Yes, this is possible, but in practice we observe this stability constraint to offer good explanations. In Table 1, and as explained in the Introduction, PSEM-4 drifts to realistic explanations with CEM-PP representing what unrealistic explanations could be. Furthermore, in Table 3, we observe exactly what you point out for the Input Reduction method on Document 2 where the explanation converges to No Words Selcted, while PSEM learns a 7 word practical explanation.
> > >
> > > > Regarding hyperparameter tuning
> > >
> > > We confirm that decisions were made manually after tuning over various combinations. In the Appendix, we note the order of which hyperparameters we focused on in order to reduce the space of all combinations. In practice, this process only need be done once and then can be used on an arbitrarily large number of explanations.
> > >
> > > > In the eq. 1 optimization problem, the objective term $f_\kappa$ seems redundant which the constraint in (c). Have the authors thought about to what extent each of these is necessary? Ideally there would be an ablation to justify the choice to keep both.
> > >
> > > We agree and have removed constraint (C) while adding penalties for all $i$ to the objective in eq. (1), specifically writing the objective as $c\sum_{i=1}^Nf_{\kappa}(x_0,\delta_i)+ \beta \sum_{i=1}^N\|\delta_i\|_1$. A sentence after the equation was added regarding this loss term. The subproblems in Algorithm 1 correspond to this formulation. Thank you for the suggestion.
> > >
> > > > So why not do this for individual pixels? Is it possible that superpixels serve another purpose that the authors aren't acknowledging here, such as mitigating adversarial or discontinuous solutions? If so, it would be best to explain that, and discuss why you don't explore other approaches, for example the total variation penalty adopted in Fong et al., 2017.
> > >
> > > We use superpixels to mitigate discontinuous solutions which is motivated by Luss et.al. (2021). We acknowledge that there are other ways to mitigate such solutions as given in the other literature such as a total variation penalty that you point out in Fong et.al. 2017. and their follow up in 2019. Stability penalties and a path could be derived from those works as well, but that is outside the scope of our paper. Our goal is to show that stability penalties can help derive paths for sufficient explanations. We clarify in Section 5.2 of the revision that sufficient explanations can be learned with different regularizations such as those in the literature you refer to.

---

> > > > ### Author Response · Authors · 2024-09-25
> > > > **Response to your review [4/4]**
> > > >
> > > > > The section 4 experiments are nearly a sanity check, because 2/3 criteria are built into the proposed method.
> > > >
> > > > These experiments also motivate what the PSEM path offers over other path algorithms. This is not just a sanity check because we must show that these characteristics are not implied for other path algorithms. It shows where other path algorithms would suffer from stability problems. Furthermore, it showcases the tradeoff of stability and fidelity for the LIME methods when comparing Table 2 with Table 6 of the Appendix, as noted at the end of Section 4.
> > > >
> > > > > Why does the Newsgroups experiment use such a simple model?
> > > >
> > > > We do not apply PSEM to LLMs, because, to the author's knowledge, the only way to learn sufficient explanations for an LLM would involve combinatorial optimization, i.e., explicitly searching over all combinations of words in a text. While saliency methods are possible for an LLM, it is not clear how to optimize of the text embeddings for an LLM. For a saliency method, such as integrated gradients, one can compute the attribution for each element of the embeddings of each token, i.e., for a  word "cat" that lies in 768 dimensional embedding space, the gradient can be computed with respect to each dimension of that embedding and an average over the 768 attributions can be used as the attribution for "cat". But for pertinent positives, one cannot simply optimize over those embeddings because one would move in a direction that likely leads to an embedding that isn't an actual word.
> > > >
> > > > > Regarding the user study
> > > >
> > > > We included a new screenshot of the instructions seen by participants in the Appendix Section C of the revision. The goal of the survey is to see if PSEM offers user's useful information. As we state in the paper, "While one can debate whether this question is biased because PSEM offers more information, it is worth noting that participants that preferred PSEM did much better on PSEM questions than those that preferred LIME or CEM-PP."
> > > >
> > > > > Lime is a strange choice of comparison} As noted in the paper, LIME is the most widely used attribution method in explanation literature, which is why we compare with it. Most of the literature discussed above was applied solely to images, and specifically tailored for color images which require a mask rather than optimizing the pixels of a grayscale image.
> > > >
> > > > Explanations for color images is only one of the modalities that we consider. Furthermore, much of the literature above is not relevant for comparison because, as noted, they assume training data. For the remaininng perturbation literature, we  acknolowedge that path algorithms could be developed for different sufficient explanation formulations in the Related Work Section 2 of the revison, however, it is out of the scope of this paper to figure out which of these frameworks is the best for sufficient explanations. For example, we already know that Fong et.al. (2019) is likely preferable to Fong et.al. (2017) which it built on and that Fong et.al. (2017) is very similar to Chang et.al (2019).

---

> > > > > ### Comment · Reviewer_Evvb · 2024-10-17
> > > > > **Response**
> > > > >
> > > > > Thanks to the authors for their response. Regarding the missing citations, there are indeed some differences between these methods, including the optimization (learned by a network vs separately for each prediction), the objective (whether to maintain or change the prediction), and the feature perturbation approach. My point in bringing up these works is 1) this is a densely explored space and it would be best to mention connections with prior works, 2) these choices are often interchangeable, which means we can instantiate new algorithms by combining them almost arbitrarily (e.g., you could learn the perturbations from "Real time image saliency for black box classifiers" on a per-prediction basis like in "Interpretable explanations of black boxes by meaningful perturbation"; your method could swap its objective to changing rather than maintaining the prediction; "Interpretable explanations of black boxes by meaningful perturbation" could parameterize perturbations using the Gumbel Softmax). Many of the ingredients in this paper's proposal exist in prior works, so it seems like the new idea here is finding a sequence of perturbations subject to the "stability" penalty.
> > > > >
> > > > > Although reasonable, this idea is pretty straightforward. The experiments also didn't strongly convince me of its utility, as I mentioned in my review regarding the user study and qualitative results for Newsgroups, CelebA and MNIST. I'll keep my assessment as-is.
> > > > >
> > > > > (Re: the Newgroups experiment: you don't need to use a "LLM", you can use a small BERT model if that makes the experiment easier. DistilBERT-Base is on Huggingface and has 70M params, it should be quick to fine-tune, and if your method doesn't work on that then I'm not sure you should claim utility for text classification.)

---

> > > > > > ### Author Response · Authors · 2024-10-17
> > > > > > **A counterfactual objective function is not interchangeable with sufficiency for path explanations**
> > > > > >
> > > > > > Thank you for the continued discussion. We would like to offer the following further responses:
> > > > > >
> > > > > > > Regarding the missing citations, there are indeed some differences between these methods, including the optimization$\ldots$
> > > > > >
> > > > > > We first note that all suggested citations have been included in the Related Work Section 2 in the revision that was submitted. This includes putting the citations into the perspective of the paper as well as making the connections as we noted "These different perturbation-based methods are extendable
> > > > > > to our framework, as will be demonstrated on a colored image dataset specifically for the method of Luss et al. (2021). ".
> > > > > >
> > > > > > > your method could swap its objective to changing rather than maintaining the prediction
> > > > > >
> > > > > > We dispute this claim for the following reasons:
> > > > > >
> > > > > > 1) Counterfactuals seeks a **minimal** change to flip a label whereas sufficiency seeks a **maximal** change to maintain the label. It is intuitive to seek a path of small changes that result in a maximal change, but this is not true for seeking the minimal change for a counterfactual as the first (minimal) change that produces a valid counterfactual is the solution.
> > > > > >
> > > > > > 2) It is not clear what such a formulation would entail as the objective of each intermediate solution would not be seeking a counterfactual as it has already been found and which should be the objective of the final sample on the path.
> > > > > >
> > > > > > 3) The Discussion Section 7 of the revision discusses future directions to consider a path of semi-factual explanations, which generalizes pertinent positives, but as noted, it is not obvious, even in that setting, how to formulate the removal of features.
> > > > > >
> > > > > > > it seems like the new idea here is finding a sequence of perturbations subject to the "stability" penalty.
> > > > > >
> > > > > > We agree. And this is stated as the first item in our list of contributions: "We propose a novel (constrained) formulation to learn a stable sequence of sufficient explanations." We maintain that this contribution is tailored to sufficient explanations for the reasons noted above.
> > > > > >
> > > > > > > you can use a small BERT model if that makes the experiment easier
> > > > > >
> > > > > > A limitation here is due to the base explanation method used for the path (not our proposal) - BERT models require optimizing over embeddings to identify which words to remove. It is, however, possible to use averages over embeddings as done for individual attribution methods like integrated gradients, although it is not very natural for a path explanation.
> > > > > >
> > > > > > > Regarding: as I mentioned in my review regarding the user study
> > > > > >
> > > > > > Your original review claimed that the user study "is perhaps the most compelling evaluation approach in the paper" so we thought the utility of the user study was understood. Please see our original response [4/4] to the issues regarding the user study that you asked about.

---

> > > > > > > ### Comment · Reviewer_Evvb · 2024-10-17
> > > > > > > **Response**
> > > > > > >
> > > > > > > **Citations:** thanks for adding the citations, that's an improvement to the paper.
> > > > > > >
> > > > > > > **Changing the objective:** perhaps I'm missing something, but this seems like a straightforward modification. Consider the feature removal approach to perturbation that you use with images. In your current problem formulation, you find a sequence of perturbations removing progressively more features, each of which maintains the original prediction. In a counterfactual version that swaps the objective to changing rather than maintaining the prediction, you would simply find a sequence of perturbations beginning from the removal of all features, removing progressively fewer features, and each of which achieves a different (perhaps user-specified) prediction. Intuitively, the features that stay removed strongly induce the original class, while the ones that get added back don't. Is there anything wrong with this? Anyway, I'm not asking that you implement this to improve the paper, I'm just pointing out that the path idea (like many design choices in this space) can be used quite flexibly in different algorithms.
> > > > > > >
> > > > > > > **BERT:** you sound convinced your approach can't work with BERT. That's unfortunate, because it's a standard technique for high-performance text classification. If your algorithm only works for bag-of-words + MLP, that's quite limiting.
> > > > > > >
> > > > > > > Up to you whether to explore this, but I think making this type of method work for BERT is actually quite straightforward. The whole point of BERT/MLM pre-training is to predict missing words represented by a mask token, so you could use that token to represent word removal within a sentence... I'm pretty sure this has been done in plenty of papers by now. A couple considerations about that are:
> > > > > > >
> > > > > > > 1. You might be concerned that the model's recognition of the mask token gets broken during task-specific fine-tuning. If so, one way to mitigate this is by incorporating random token masking during fine-tuning; in fact, this is advocated for in several interpretability papers that rely on feature removal, even for non-BERT classifiers (can provide references if helpful).
> > > > > > >
> > > > > > > 2. You might still be concerned about optimization and whether you have to rely on combinatorial searches. I agree that this would be cumbersome, although a greedy forward/backward selection sounds like a reasonable first approach. If you want a gradient-based optimization procedure, you could parameterize each token's removal as a binary random variable, and optimize over it either 1) in a continuous fashion (sort of like in "Interpretable explanations of black boxes by meaningful perturbation"), or 2) with a binary Gumbel Sigmoid ($\approx$Bernoulli random variable).

---

> > > > > > > > ### Author Response · Authors · 2024-10-17
> > > > > > > > **Thank you for the interesting suggestions**
> > > > > > > >
> > > > > > > > > Changing the objective
> > > > > > > >
> > > > > > > > Counterfactuals are typically defined as a perturbed sample where the label has changed and the original sample and perturbed sample are minimally different. See "Explaining NLP Models via Minimal Contrastive Editing (MICE)", ACL-IJCNLP (2021) or "Let the CAT out of the bag: Contrastive Attributed explanations for Text", EMNLP (2022) which use expensive combinatorial optimization to find counterfactuals in classification models. These problems are typically solved by starting at the original sample because a minimal change is desired.
> > > > > > > >
> > > > > > > > While your suggestions are interesting, we offer the following comments:
> > > > > > > >
> > > > > > > > 1) It would be expensive to find features to add back that maintain a particular class that are not related to the original class.
> > > > > > > >
> > > > > > > > 2) Adding features back would likely jump between different counterfactual classes which would take away insight from any such path.
> > > > > > > >
> > > > > > > > 3) The blank sample you suggest starting with could be predicted the same as the original sample, since the classifier needs to make some prediction on it which will bring up the issue of where to start the search.
> > > > > > > >
> > > > > > > > > Regarding BERT
> > > > > > > >
> > > > > > > > Your suggestions are again interesting. Such masking procedures are how individual counterfactuals have been learned in the literature cited above (using expensive combinatorial optimization). In theory, one could apply masking and combinatorial optimization to discover sufficient explanations as well, however this may not scale as well for a path of explanations as gradient optimization is much faster. We will add such suggestions in the Discussion section in the final version.
> > > > > > > >
> > > > > > > > In summary, we greatly appreciate that there are other possible path algorithms to try, as you are suggesting. For attribution methods, we have implemented some for comparison in Section 4 for our quantitative evaluations. We hope this paper positions the idea that a path of explanations can offer users more insight. We certainly do not mean to preclude researchers from expanding with other ideas for path algorithms.

---

### Review · Reviewer_CGdQ · 2024-07-26

**Summary Of Contributions:**

The paper proposes a novel method called Path-Sufficient Explanations Method (PSEM),
which is consistent with several criteria for XAI, specifically fidelity, stability, and comprehensibility,
and also overcomes some limitations of existing methods (e.g., CEM-PP and LIME).

Given a local explanation problem comprising a classifier and an input,
the proposed PSEM will output a sequence of stable, sufficient, and monotonic (on size or value) explanations starting from the original input to a subset-minimal sufficient input.
This path as a whole serves as the explanation for the given input, making it more informative.
In contrast, existing XAI methods, including LIME and CEM-PP, can only show a single explanation.
Although LIME and CEM-PP can be adapted to produce a path of explanations, they face some problems:

1. For CEM-PP, one major problem is that its explanation path may lack stability; that is, a sufficient explanation at the $i$-th iteration may have a size larger than a sufficient explanation at the $(i-1)$-th iteration, which is counterintuitive (see Fig. 1).

2. For LIME, one major problem is that its explanation may lack fidelity; that is, the computed explanation may result in different predictions than the original prediction (see Tab. 2). It is true that LIME actually targets positively relevant and negatively relevant features rather than sufficiency.

The conducted evaluations (both quantitative and qualitative) and the user study suggest that
users would have greater trust in PSEM explanations.
Specifically, the user study conveyed that the additional information from PSEM does not overwhelm but rather significantly improves understanding.

**Audience:**

Yes

**Broader Impact Concerns:**

No concerns.

**Claims And Evidence:**

Yes

**Requested Changes:**

Some questions:

1. Is the hyperparameter $N$ (the length of the PSEM path) predefined, or is it determined by the search algorithm?

2. Is it true that the order of removing features impacts the $i$-th sufficient explanation, and therefore the PSEM path?
If this is the case, then there may exist more than one PSEM path. This suggests that it is possible to enumerate PSEM paths, which may provide more valuable information regarding the classifier.
Can the author comment on this aspect?

3. There are papers studying the enumeration of sufficient explanations (even though their methods differ from the proposed one):
Alexey Ignatiev, Nina Narodytska, Nicholas Asher, Joao Marques-Silva. From Contrastive to Abductive Explanations and Back Again. AI*IA-2020.
Adnan Darwiche, Chunxi Ji. On the Computation of Necessary and Sufficient Explanations. AAAI-2022.
It would be interesting to see if the proposed method can compute more than one PSEM path for the same input.


Typos:
1. P5. "The third constraint in (a) represents feasibility for the domain." There is no third constraint in (a).
2. P6. "...note that CEM-PP is most..." => "...CEM-PP is the most.."
3. P8. "Results on ten images are shown in ..." => "The results on ten ..."

**Strengths And Weaknesses:**

Strengths:

1. The paper is interesting, easy to follow, with clear motivation.
2. The proposed method overcomes some limitations of existing XAI methods, thus the contribution of the paper is clear.
3. The experiments include both text and image data, which are to some extent comprehensive.
4. Code is provided in the supplementary material, so reproducibility is not a concern.


Weaknesses:

1. Although the proposed idea is interesting, the novelty of the proposed method is limited.

---

> ### Author Response · Authors · 2024-09-25
> **Response to your review**
>
> Thank you very much for your favorable comments and giving our paper your attention. We are glad that you appreciate the ideas and contributions our paper makes as demonstrated by your summary of the strengths. Regarding the novelty, we view our contribution to be one of insight into creating path explanation methods (as well as showing which types of path methods could benefit). While we focus on a path of sufficient explanations as defined by pertinent positives, our framework could be adjusted to learn paths for pertinent negatives. The formulation for a path of semi-factual explanations (as suggested by Reviewer vf3N) is not clear but is an interesting direction to think about. We added comments on this aspect in the Discussion Section 7 of our revision.
>
> Below we also address your questions.
>
> > Is the hyperparameter $N$ (the length of the PSEM path) predefined?
>
> Yes, we take $N$ as an input that must be tuned (this is what we meant by deeming it a hyperparameter rather than parameter but we will clarify in the paper). As we note in Appendix A, while hyperparameter tuning can be extensive, it is important to consider that, for a given dataset and application, the parameters need only be tuned once, after which, an arbitrary number of predictions can be explained.
>
> > Is it true that the order of removing features impacts the $i$-th sufficient explanation, and therefore the PSEM path?
>
> Yes, and this is because sufficient explanations are not necessarily unique. Take for example a symmetric image such as the 1 in Figure 3. The PSEM path begins by removing pixels from the top of the 1, but an equally minimal sufficient explanation might have removed pixels from the bottom of the 1 instead (of course also depending on the classifier). Similar situations could occur for a loan applicant where two different features have similar effects on the classifier. Then reducing one over the other could have different interactive impacts on the other features. An enumeration of PSEM paths could lead to an interesting direction (with ideas pursuing that addressed in the next response).
>
> > It would be interesting to see if the proposed method can compute more than one PSEM path for the same input.
>
> Two different paths could be obtained, for example, by using different $\beta$ values at the first iteration. Two different PSEM-1 explanations would be learned, one sparser than the other, but not necessarily connected by an intermediate sufficient explanation for the same reasons as discussed in the paper. From these two PSEM-1 explanations, different PSEM paths could be learned. Another idea could be to start from different initial points which could lead to different local minima. We have added comments on the non-uniqueness of the solution to the end of Section 3 in the revision. Deriving two different PSEM paths with similar sparsity would require there exist two sufficient solutions with identical prediction scores. Thank you for the literature that we have added in the Related Work Section 2 of the revision. While these works address explanations for logic-based models, e.g., decision graphs, and rely on discrete searches, the idea of searching for multiple paths is interesting as you suggest, and we have also added this direction to the end of the Discussion Section of the revision.

---

### Review · Reviewer_vf3N · 2024-09-20

**Summary Of Contributions:**

The authors leverage three key criteria in explainable artificial intelligence (XAI), viz. fidelity, stability, and comprehensibility, and apply them to feature attribution methods to generate minimally sufficient explanations to explain the decision of the underlying model for a given input. The proposed work integrates the aforementioned properties to propose the Path-Sufficient Explanations Method (PSEM), which outputs a sequence of sufficient explanations of strictly decreasing size. Qualitative and quantitative results across three modalities (image, tabular, and text) show the benefits of generating sufficient paths of explanations.

**Audience:**

Yes

**Broader Impact Concerns:**

Not applicable.

**Claims And Evidence:**

Yes

**Requested Changes:**

Please address the above open questions in the weakness section.

**Strengths And Weaknesses:**

**Strengths**

1. The authors perform extensive qualitative and quantitative experiments to show the effectiveness of PSEM on tabular, text, and vision datasets.

2. The proposed Path-Sufficient Explanations Method generates a path of sufficient explanations that highlight the evolution of the final explanations under different fidelity, feasibility, and stability criteria.

**Weaknesses/Open Questions**

1. Is it fair to compare PSEM with LIME as it tries to fit a regression model and doesn't have an explicit constraint like PSEM to ensure that the predicted labels are consistent with the original predictions? Moreover, the LIME explanations are generated by fitting a regression model on multiple classes and there are no constraints that force the optimization to generate faithful explanations like in PSEM.

2. The use of MNIST for user studies is unclear as the ground-truth logic for classifying digits is subjective. Wouldn't it make more sense to test it on CelebA datasets where users know and probably agree on how to classify attributes like smile vs no smile?

3. The formulation of PSEM is very similar to generating semi-factual explanations with some added constraints and questions about the novelty of the proposed framework.

4. While the author defines stability as the % of features at each index (> 1) along a path that also appeared at the previous index, the interpretation of this metric is very different than the formal definition of stability in XAI literature.

5. Minor: *We trained a three-layer neural network that achieves 64% test accuracy.* -----> While the authors argue that their goal is to explain model predictions and not build a state-of-the-art model, the choice of the model achieving sub-par accuracy is unclear in the current experimental setup as multiple model variants could lead to improved performance on the downstream task.

---

> ### Author Response · Authors · 2024-09-25
> **Response to your review**
>
> Thank you for your feedback. We are glad that you found our experiments to effectively demonstrate the algorithm. We address your major questions below:
>
> > Is it fair to compare PSEM with LIME as it tries to fit a regression model and doesn't have an explicit constraint like PSEM to ensure that the predicted labels are consistent with the original predictions?
>
> Great question! We compare with LIME because it is the most common attribution method used in XAI literature (mentioned in the paper). Without an explicit constraint as with PSEM, we should not expect it to compete in terms of fidelity, but it is important to verify that LIME lacks this property naturally, in order to justify the benefit of explicitly including this constraint. As also noted in the paper, CEM-PP is thus clearly the most relevant explanation method for comparison, where sufficiency still requires the constraint.
>
> Regarding the multiple classes, LIME computes independent regressions for each class. In terms of explainability, we are interested in explaining the decision of the model, i.e., explaining the predicted class of the model, and this is the class being explained in the experiments. The question about fidelity is whether this local approximation can be used to make predictions that agree with the predictions of the model being explained.
>
> > The use of MNIST for user studies is unclear as the ground-truth logic for classifying digits is subjective. Wouldn't it make more sense to test it on CelebA datasets where users know and probably agree on how to classify attributes like smile vs no smile?
>
> We contend that the logic for classifying digits is actually intuitive as judged by the extremely high performance attainable on this task with relatively small convolutional models (as the importance for predictability is a function of nearby pixels). We noted in Section 6 "while MNIST might appear simplistic, the task for participants described below is not easy, as witnessed by the accuracies in Figure 5." Indeed, we used MNIST because we thought users also probably agree on how to classify a 1 vs 3, etc. As noted in the paper in Section 6, we used MNIST instead of CelebA because ``color images use an extended model that is not as widely applicable as the first model via (2)". Moreover, the reason for such a model to classify someone as smiling might be very different than what we might intuitively surmise, much more so than with digits given the added the complexity associated with colored facial images with latent concepts such as presence/absence of makeup etc. (Luss et. al. KDD 2021).
>
> > The formulation of PSEM is very similar to generating semi-factual explanations with some added constraints and questions about the novelty of the proposed framework.
>
> Semi-factual explanations are (subtly) different than even pertinent positive explanations (also known as abductive explanations). Semi-factual explanations allow features to move in any direction whereas pertinent positives constrain the change to a reduction. Our contribution is learning a path of explanations which often results in a better explanation than trying to sparsify the explanation directly as with CEM-PP. The monotonicity of the pertinent positive explanations is crucial to the path because it is clear how to formulate the monotonic removal of features. As such, ours is a meta-approach, where the base explanation could be pertinent positives or semi-factuals. Although, in the latter case it is not obvious how to formulate a general (smooth) removal of features, and is an interesting future direction that we have added to the Discussion Section 7.
>
> > ..., the interpretation of this metric is very different than the formal definition of stability in XAI literature.
>
> We agree that stability is defined differently in XAI literature but this definition pertains to individual explanations like LIME or CEM-PP, where stability measures that the explanation is similar for similar samples. We are specifically defining a measure of stability for a path of explanations. We have added comments clarifying the difference in Section 4.

---

### Author Response · Authors · 2024-09-25
**General comment for all reviewers**

Thank you to all reviewers for your time and feedback. We have addressed all your comments below, as addressed to you each individually, and hope we have allayed your concerns. We have also submitted a revision where new text appears in blue to make it clear. We point to where changes have been made in our corresponding responses.

---

### Decision · Action_Editor_z9Gb · 2024-12-01

**Recommendation:** Accept as is

**Comment:**

While two reviewers were quite negative about the paper, I read the updated manuscript and their main critiques have been indeed addressed by the authors.  My take is that any additional experiments (e.g., extensions to BERT) could very well be addressed in follow-up work.  Also, the literature review was improved post-rebuttal.

**Audience:**

Definitely.  XAI and practical algorithms for sufficient explanations are very much within the scope of TMLR.

**Claims And Evidence:**

The paper introduces an improved algorithm for identifying (monotonic sequences of) sufficient explanations of model decisions.  The claims that the algorithm does a better job at implementing useful desiderata (specifically, stability and fidelity) seem to be supported by both the optimization problem and the empirical comparison.  The latter considers two reasonable competitors (one, LIME, is not very recent, but it still is a good baseline) and three different data modalities.  A user study further supports the claim.